# Unifying Proportional Fairness in Centroid and Non-Centroid Clustering

**Benjamin Cookson**
Department of Computer Science
University of Toronto
bcookson@cs.toronto.edu

**Nisarg Shah**
Department of Computer Science
University of Toronto
nisarg@cs.toronto.edu

**Ziqi Yu**
Department of Computer Science
University of Toronto
ziqiyu@cs.toronto.edu

## Abstract

Proportional fairness criteria inspired by democratic ideals of proportional representation have received growing attention in the clustering literature. Prior work has investigated them in two separate paradigms. Chen et al. [1] study *centroid clustering*, in which each data point's loss is determined by its distance to a representative point (centroid) chosen in its cluster. Caragiannis et al. [2] study *non-centroid clustering*, in which each data point's loss is determined by its maximum distance to any other data point in its cluster.

We generalize both paradigms to introduce *semi-centroid clustering*, in which each data point's loss is a combination of its centroid and non-centroid losses, and study two proportional fairness criteria—the core, and its relaxation, fully justified representation (FJR). Our main result is a novel algorithm which achieves a constant approximation to the core, in polynomial time, even when the distance metrics used for centroid and non-centroid loss measurements are different. We also derive improved results for more restricted loss functions and the weaker FJR criterion, and establish lower bounds in each case.

## 1 Introduction

Clustering is a central task in AI, in which the goal is to either partition a set of $n$ data points into $k$ clusters $C_1, \ldots, C_k$, or return $k$ centroids $x_1, \ldots, x_k$, or perhaps return both the clusters and their assigned centroids. In many cases, partitioning the data points and choosing centroids are viewed as interchangeable. For example, in the traditional clustering setting, the popular $k$-means loss function (to be minimized) can be written in two equivalent forms:

$$\min_{C_1,\ldots,C_k} \sum_{t=1}^{k} \frac{1}{|C_t|} \sum_{i,j \in C_t} d(i,j)^2 \equiv \min_{C_1,\ldots,C_k,x_1,\ldots,x_k} \sum_{t=1}^{k} \sum_{i \in C_t} d(i,x_t)^2.$$

This is because for a given partition $(C_1, \ldots, C_k)$, the optimal choice of the centroid $x_t$ for each cluster $C_t$ is the average of its points (this can be generalized to non-Euclidean metrics too) and plugging in these optimal choices yields the formulation on the left.

An emerging line of research at the intersection of economics and AI focuses on applications in which every data point represents an agent, and studies facets such as dealing with strategic manipulations

39th Conference on Neural Information Processing Systems (NeurIPS 2025).

by selfish agents [3–7] or ensuring fairness to the agents [8–10]. Here, it is not sufficient to have only a global loss function to be minimized; one needs to know the loss functions (i.e., preferences) of the individual agents and different formulations of loss functions at the agent level, even if equivalent in the aggregate, lead to different results.

Our focus is on fairness—specifically, on *proportional fairness*. Informally, it provides a fairness guarantee to every possible group of agents—not just those determined by a limited set of sensitive attributes such as race and gender, or their intersections—with the strength of the guarantee scaling proportionally to the group's size and cohesiveness of its members' preferences. Proportional fairness criteria have been applied to a growing number of problems in AI [1, 9–13].

For clustering, Chen et al. [1] initiate the study of proportional fairness for *centroid clustering*, in which the loss of an agent is its distance to the centroid of its cluster. They motivate it through facility location, in which the goal is to build $k$ facilities; each agent naturally prefers to be close to its assigned facility. Caragiannis et al. [2] study proportional fairness for *non-centroid clustering*, in which the loss of an agent is an aggregate (e.g., average or maximum) of its distances to the other agents in its cluster. Their motivation stems from clustered federated learning, in which data sources (agents) are partitioned into clusters and agents in each cluster collaboratively learn a model; each agent then wishes to be close (in terms of their data distributions) to the other agents in its cluster, so that the model they collaboratively learn is accurate for the agent personally.

But in both these applications, and in many others, it may be natural for agents to care about *both* their distance to the centroids of their clusters and their distances to the other agents in their clusters. For example, in facility location, an individual may want their assigned facility to be close to their house as well as other individuals using the same facility to also be from their neighborhood, which makes it more likely to bump into friends and acquaintances. In clustered federated learning, the centroids may represent the models collaboratively learned by the different clusters of agents. An agent may care about both the model being effective for their own data distribution and the other agents in their cluster having data distributions similar to their own, as such cohesion can be important to inducing trust and cooperation within the cluster.

It may even be the case that agents' preferences are formed out of a combination of centroid and non-centroid losses that are induced by completely different metric spaces. When a teacher groups students together for final class projects, students' preferences for which group they are placed in would depend on the project topic that group is assigned (centroid) and which other students are in that group (non-centroid). Two students may be very close friends but have completely different opinions on project topics.

This motivates us to initiate the study of proportional fairness for a natural generalization of centroid and non-centroid clustering that we term *semi-centroid clustering*, in which the loss of each agent $i$ is a function of both the cluster of agents $C_t$ it is a part of (i.e., $i \in C_t$) and the centroid $x_t$ assigned to this cluster.

By leveraging approaches common in the works of Chen et al. [1] and Caragiannis et al. [2], along with novel and intricate techniques we develop, we are able to attain desirable proportional fairness guarantees for semi-centroid clustering.

## 1.1 Our Results

| | Dual Metric Loss | | Weighted Single Metric Loss | |
|---|---|---|---|---|
| | Core | FJR | Core | FJR |
| Existential Upper Bound | 3 | 1 | $\min\{2/\lambda, 3\}$ | 1 |
| Efficient Upper Bound | $3 + 2\sqrt{3}$ | 4 | $\min\{2/\lambda, f_\lambda\}$ | $\min\{4, 2/\lambda, f_\lambda\}$ |
| Lower Bound | 2 | 1 | $\max\{g_\lambda, \frac{2(1-\lambda)}{2\lambda+1}\}$ | 1 |

Table 1: Core and FJR approximations for semi-centroid clustering under dual metric loss and weighted single metric loss. Here, $f_\lambda = \frac{\sqrt{2\lambda - 11\lambda^2 + 13} + 3 - \lambda}{2 - 2\lambda}$ and $g_\lambda = \frac{\sqrt{\lambda^2 - 2\lambda + 5} - \lambda + 1}{2}$.

A semi-centroid clustering algorithm takes a set of $n$ data points (agents) as input and returns both a partition $(C_1, \ldots, C_k)$ of the data points and the corresponding centroids $(x_1, \ldots, x_k)$. The loss of each agent $i \in C_t$ is measured by a loss function $\ell_i(C_t, x_t)$, which depends both on the cluster $C_t$ it belongs to and its centroid $x_t$. While some of our results apply to arbitrary loss functions, most focus on two structured families. Dual metric loss measures the sum of agent $i$'s distance to its centroid $x_t$ according to a metric $d^c$ and its maximum distance to any agent $j \in C_t$ in its cluster according to a different metric $d^m$. Weighted single metric loss is the special case in which both metrics are scaled versions of a common metric, i.e., $d^c = (1 - \lambda) \cdot d$ and $d^m = \lambda \cdot d$ for some metric $d$ and $\lambda \in [0, 1]$.

We investigate (multiplicative approximations of) two proportional fairness criteria studied in prior work on centroid and non-centroid clustering: the core, and its relaxation, fully justified representation (FJR). Intuitively, when forming $k$ clusters given $n$ data points, the core demands that there is no set $S$ of at least $n/k$ agents, who *proportionally deserve* to be able to form a cluster, and a feasible centroid $y$ such that every member of $S$ prefers $(S, y)$ to its currently assigned cluster. FJR puts a greater demand on successful violations: even the maximum loss of any agent $i \in S$ for $(S, y)$ should be lower than the minimum loss of any agent $i \in S$ under the algorithm's clustering.

First, we show that none of the algorithms developed in prior work for centroid or non-centroid clustering work in our more general paradigm of semi-centroid clustering, even for the restricted family of weighted single metric loss. Then, for both loss families, we obtain core and FJR approximation guarantees, both existential and polynomial-time attainable, by designing novel algorithms, and prove (existential) lower bounds. Our main results are summarized in Table 1.

Finally, we also evaluate the performance of our algorithms on real-world datasets in Appendix F. We observe that our algorithms consistently achieve near-perfect approximations of the fairness notions we consider, beating both the theoretical worst-case bounds we establish, and the performance of classical clustering algorithms such as k-means++. We also show that they achieve these guarantees while sacrificing classical clustering objectives only slightly. This generalizes similar observations for centroid and non-centroid clustering in prior work [1, 2].

## 1.2 Related Work

As mentioned above, there are several existing papers studying proportional fairness in clustering. Chen et al. [1] are the first to introduce the notion of the core for *centroid* clustering, and use the *Greedy Capture* algorithm to achieve a $(1 + \sqrt{2})$-approximation to the core. Micha and Shah [14] improve the approximation factor to 2 for the Euclidean $L^2$ metric in the "unconstrained setting" where the centroids can be placed anywhere in $\mathbb{R}^t$. Aziz et al. [15] observe that in this unconstrained setting, instead of the infinitely many possible centroid locations, one can focus on just the $n$ agent locations and achieve a constant approximation to the core in polynomial time. Li et al. [16] study a stronger criterion than the core. Caragiannis et al. [2] study proportional fairness in *non-centroid* clustering, and use a variation of the Greedy Capture algorithm to achieve 2-core with respect to the maximum-distance loss. They also study FJR as a weakening of the core. Ebadian and Micha [17] use the centroid clustering framework to study the problem of sortition. In this setting, they make the set of possible centroids equal to the set of agents, and look for a lottery over centroid clusterings with the property that each agent has an equal probability of being selected as a centroid in the chosen clustering. They achieve such a lottery with ex-post proportional fairness guarantees using an algorithm inspired by Greedy Capture.

Li et al. [18] study a similar graph-based problem of partitioning friends into groups, which is equivalent to non-centroid clustering, albeit when each person has a utility (rather than a loss) of 1 for every friend of theirs placed in their cluster. Arkin et al. [19] look at the multi-dimensional geometric stable roommates problem, which is equivalent to the non-centroid clustering model of [2], but focus on special cases such as $k = n/3$.

Finally, other fairness notions have also been investigated for clustering (see the survey by Chhabra et al. [20]). The most prominent one assumes that every data point belongs to one of many classes, and certain protected classes must have equal representation across the clusters [21, 22].

## 2 Preliminaries

### 2.1 Clustering Model

A *semi-centroid* clustering instance is a tuple $(\mathcal{N}, \mathcal{M}, \{\ell_i\}_{i \in \mathcal{N}}, k)$, where $\mathcal{N}$ is a set of $n$ agents; $\mathcal{M}$ is a set of possible cluster centers; $\ell_i : \{C \subseteq \mathcal{N} : i \in C\} \times \mathcal{M} \to \mathbb{R}_{\geqslant 0}$ is the loss function of agent $i \in \mathcal{N}$, with $\ell_i(C, x)$ being the loss incurred when it is part of cluster $C$ and assigned cluster center $x$; and $k \in \mathbb{N}$ is the number of clusters to be returned by the algorithm.

A clustering algorithm takes such an instance and returns a clustering $\mathcal{X} = \{(C_1, x_1), \ldots, (C_k, x_k)\}$, where $(C_1, \ldots, C_k)$ is a disjoint partition of the set of agents $\mathcal{N}$ (so, $\cup_{t \in [k]} C_t = \mathcal{N}$ and $C_t \cap C_{t'} = \emptyset$ for all distinct $t, t' \in [k]$). We refer to each tuple $(C_t, x_t) \in \mathcal{X}$ as a *cluster*, with $C_t$ as the *member set* (and the agents in $C_t$ as the *members*) of the cluster and $x_t$ as the *center* assigned to the cluster.

Given a clustering $\mathcal{X}$ and an agent $i \in \mathcal{N}$, let $\mathcal{X}(i)$ denote the (unique) index of the cluster of which $i$ is a member (i.e., $i \in C_{\mathcal{X}(i)}$). The loss incurred by each agent $i \in \mathcal{N}$ under clustering $\mathcal{X}$ is $\ell_i(C_{\mathcal{X}(i)}, x_{\mathcal{X}(i)})$, which, with slight abuse of notation, we also write as $\ell_i(\mathcal{X})$ for brevity.

### 2.2 Loss Functions

Some of our results hold for arbitrary loss functions as defined above. However, the more interesting results are obtained for two structured families of loss functions that naturally combine previously-studied loss functions for centroid and non-centroid clustering.

**Dual metric loss**: We are given two distance metrics,[1] a *centroid metric* $d^c : \mathcal{N} \times \mathcal{M} \to \mathbb{R}_{\geqslant 0}$ and a *non-centroid metric* $d^m : \mathcal{N} \times \mathcal{N} \to \mathbb{R}_{\geqslant 0}$. Given a clustering $\mathcal{X}$ and an agent $i \in \mathcal{N}$, the centroid metric induces its *centroid loss* $\ell_i^c(\mathcal{X}) \triangleq d^c(i, x_{\mathcal{X}(i)})$, which measures the distance of agent $i$ to its assigned center, and the non-centroid metric induces its *non-centroid loss* $\ell_i^m(\mathcal{X}) = \max_{j \in C_{\mathcal{X}(i)}} d^m(i, j)$, which measures the maximum distance of agent $i$ to any other agent in its cluster. The former is the canonical loss function used for centroid clustering [1, 14, 15, 23] while the latter is a loss function for which appealing fairness guarantees have been derived for non-centroid clustering [2].

The overall *dual metric loss* of agent $i$ is the sum of the centroid and non-centroid parts:

$$\ell_i(\mathcal{X}) = \ell_i^m(\mathcal{X}) + \ell_i^c(\mathcal{X}) = \max_{j \in C_{\mathcal{X}(i)}} d^m(i, j) + d^c(i, x_{\mathcal{X}(i)}).$$

It is worth remarking that adding scaling factors so that $\ell_i(\mathcal{X}) = \alpha \cdot \ell_i^m(\mathcal{X}) + \beta \cdot \ell_i^c(\mathcal{X})$ for some $\alpha, \beta \geqslant 0$ would not make the family any more general; one can instead just use $\widehat{d^m} = \alpha \cdot d^m$ and $\widehat{d^c} = \beta \cdot d^c$ as the new non-centroid and centroid metrics, respectively.

**Weighted single metric loss:** A special case of dual metric loss is where both centroid and non-centroid losses use (scaled versions of) the same metric $d : (\mathcal{N} \cup \mathcal{M}) \times (\mathcal{N} \cup \mathcal{M}) \to \mathbb{R}_{\geqslant 0}$. That is, for some $\lambda \in [0, 1]$, we have $d^m(i, j) = \lambda \cdot d(i, j)$ for all $i, j \in \mathcal{N}$ and $d^c(i, x) = (1 - \lambda) \cdot d(i, x)$ for all $i \in \mathcal{N}$ and $x \in \mathcal{M}$, so that the overall loss is given by

$$\ell_i(\mathcal{X}) = \lambda \cdot \max_{j \in C_{\mathcal{X}(i)}} d(i, j) + (1 - \lambda) \cdot d(i, x_{\mathcal{X}(i)}).$$

Once again, note that using arbitrary non-negative scaling factors $\alpha$ and $\beta$ instead of $\lambda$ and $1 - \lambda$ does not offer any further generalization as one can equivalently use $\lambda = \frac{\alpha}{\alpha + \beta}$.

Non-centroid and centroid clustering are easily seen as special cases of the weighted single metric loss (which is, in turn, a special case of the dual metric loss) with $\lambda = 1$ and $\lambda = 0$, respectively. Hence, the semi-centroid clustering algorithms we design for dual metric loss and for weighted single metric loss can be used for both centroid and non-centroid clustering.

### 2.3 Proportional Fairness Guarantees

We focus on two proportional fairness criteria prominently studied by prior work.

---

[1]More precisely, these are pseudometrics. A pseudometric $d : X \times X \to \mathbb{R}_{\geqslant 0}$ satisfies $d(x, x) = 0$ for all $x \in X$, $d(x, y) = d(y, x)$ for all $x, y \in X$, and $d(x, y) \leqslant d(x, z) + d(z, y)$ for all $x, y, z \in X$.

**Definition 1** (The Core). For $\alpha \geqslant 1$, a clustering $\mathcal{X} = \{(C_1, x_1), \ldots, (C_k, x_k)\}$ is in the $\alpha$-core if there is no group of agents $S \subseteq \mathcal{N}$ with $|S| \geqslant {}^n/k$ and feasible center $y \in \mathcal{M}$ such that $\alpha \cdot \ell_i(S, y) < \ell_i(\mathcal{X})$ for all $i \in S$ (if this happens, we say that coalition $S$ deviates with center $y$).

In words, no group $S$ of at least ${}^n/k$ agents, which proportionately deserves a cluster center representing it, can unilaterally improve by being a cluster of its own with a feasible center $y$.

The second criterion is a relaxation that imposes a higher bar on deviating coalitions.

**Definition 2** (Fully Justified Representation (FJR)). For $\alpha \geqslant 1$, a clustering $\mathcal{X} = \{(C_1, x_1), \ldots, (C_k, x_k)\}$ is $\alpha$-FJR if there is no group of agents $S \subseteq \mathcal{N}$ with $|S| \geqslant {}^n/k$ and feasible center $y \in \mathcal{M}$ such that $\alpha \cdot \ell_i(S, y) < \min_{j \in S} \ell_j(\mathcal{X})$ for all $i \in S$.

That is, the new loss for each member of the deviating coalition, scaled by $\alpha$, should not only be less than its own loss prior to deviation, but less than the loss of any deviating member prior to deviation. It is then clear that $\alpha$-FJR is a relaxation of $\alpha$-core for all $\alpha \geqslant 1$.

**Proposition 1.** *For every $\alpha \geqslant 1$, every clustering in the $\alpha$-core is also $\alpha$-FJR.*

Due to space constraints, proofs missing from each section of the main body are presented in the correspondingly titled section of the appendix.

## 3 Greedy Capture Algorithms

In prior work on proportionally fair clustering, the *Greedy Capture* algorithm has been the go-to way to achieve a constant approximation of proportional fairness criteria in both the centroid and non-centroid worlds. Chen et al. [1] introduce it for centroid clustering and Caragiannis et al. [2] adapt it to non-centroid clustering. The centroid and non-centroid variants, presented formally as Algorithms 2 and 3 respectively in Appendix B, share many similarities. In both variants, every agent starts off as "uncaptured" and a number of balls start growing at the same rate. As soon as a ball covers at least ${}^n/k$ uncaptured agents, it captures them and becomes "open" (and in the centroid variant, the center of the open ball is added as one of at most $k$ cluster centers to be selected by the algorithm). But the two variants differ in three key dimensions.

1. *Where the balls grow:* In the centroid version, a ball grows around every feasible cluster center. In the non-centroid version, which has no centers, one grows around each agent.

2. *Whether open balls keep growing:* In the centroid variant, a ball continues to grow after it opens, capturing any uncaptured agents as soon as it covers them. In the non-centroid variant, a ball stops growing once it opens. This is intuitive because an open ball that grows and captures an agent far away would not worsen previously-captured agents in the centroid paradigm, but would do so in the non-centroid paradigm.

3. *Whether agents switch clusters in the end:* In the centroid variant, the algorithm simply returns a set of (up to) $k$ cluster centers and each agent's loss is their distance to the nearest cluster center. Note that this may not be the center of the ball that captured it as the algorithm may have added a closer center afterwards. Hence, the set of agents captured by a ball when it opens may not form a single cluster. In the non-centroid variant, there is no "switching" in the end and the set of agents captured by a ball together do form a cluster. This is again because allowing an agent to switch to a different cluster she prefers could significantly worsen existing agents in that cluster.

Chen et al. [1] prove that centroid greedy capture yields a clustering in the $(1 + \sqrt{2})$-core in the centroid paradigm, and Caragiannis et al. [2] prove that non-centroid greedy capture yields a clustering in the 2-core in the non-centroid paradigm. This naturally raises the question: *Can some variant of greedy capture get a constant approximation to the core in the semi-centroid clustering paradigm?*

Unfortunately, we can show that solving the semi-centroid paradigm is not that simple. To see why, it is sufficient to consider the restricted case of the weighted single metric loss, in which the loss function is characterized by a single metric $d$ and a parameter $\lambda \in [0, 1]$. Note that the greedy capture algorithms only take in $d$ as input, and not $\lambda$. Thus, for (some variant of) greedy capture to achieve $\alpha$-core with respect to the weighted single metric loss (for any $\lambda \in [0, 1]$), it must achieve $\alpha$-core with respect to the non-centroid loss ($\lambda = 1$) and the centroid loss ($\lambda = 0$) simultaneously. In our first result, we show that no algorithm can achieve such a simultaneous guarantee.

**Theorem 1.** *For every $\alpha \geqslant 1$ and $\beta \geqslant 1$, there exists a semi-centroid clustering instance in which no clustering is simultaneously in the $\alpha$-core with respect to the centroid loss and in the $\beta$-core with respect to the non-centroid loss.*

## 4 The Core

Given Theorem 1, we switch our attention to rules that do not simply take a single metric $d$ as input; that is, for the weighted single metric loss, they also take $\lambda$ as input, and for the more general dual metric loss, they take both the centroid and non-centroid metrics, $d^c$ and $d^m$, as inputs.

In Section 4.1, we present results for the dual metric loss. Later, in Section 4.2, we present improvements for the restricted case of the weighted single metric loss.

### 4.1 Dual Metric Loss

---

**Algorithm 1:** Dual Metric Algorithm

---

**Input:** Set of agents $\mathcal{N}$, set of feasible cluster centers $\mathcal{M}$, non-centroid metric $d^m$, centroid metric $d^c$, number of centers $k$, growth factor $c \geqslant 1$, MCC approximation factor $\alpha$

**Output:** Clustering $\mathcal{X} = \{(C_1, x_1), \ldots, (C_k, x_k)\}$

1   Initialize: $\mathcal{N}' \leftarrow \mathcal{N}, \widehat{\mathcal{X}} \leftarrow \emptyset, t \leftarrow 1$;

    `// Phase 1: Build a tentative clustering` $\widehat{\mathcal{X}}$

2   **while** $\mathcal{N}' \neq \emptyset$ **do**                              `// Uncaptured agents remain`

      `// Find an` $\alpha$`-MCC cluster` $(\widehat{C}_t, x_t)$ `and save its maximum loss as` $r_t$

3      $(\widehat{C}_t, x_t) \leftarrow$ An $\alpha$-most cohesive cluster (of threshold $n/k$) from $\mathcal{N}', \mathcal{M}$;

4      $r_t \leftarrow \max_{i \in \widehat{C}_t} \ell_i(\widehat{C}_t, x_t)$;

      `// Add the cluster to` $\widehat{\mathcal{X}}$ `and mark its agents as captured`

5      $\widehat{\mathcal{X}} \leftarrow \widehat{\mathcal{X}} \cup \{(\widehat{C}_t, x_t)\}$;

6      $\mathcal{N}' \leftarrow \mathcal{N}' \setminus \widehat{C}_t$;

7      $t \leftarrow t + 1$;

    `// Phase 2: Selectively allow agents to switch to a better cluster`

8   $\forall t \in [k], C_t \leftarrow \widehat{C}_t$;

9   **for** $i \in \mathcal{N}$ **do**

10     $t \leftarrow \widehat{\mathcal{X}}(i)$;

      `// Clusters agent` $i$ `can switch to without harming its members too much`

11     $V_i \leftarrow \{t' \in [k] : \forall j \in \widehat{C}_{t'}, d^c(j, x_{t'}) + d^m(j, i) \leqslant c \cdot r_{t'}\}$;

      `// Among them, agent` $i$ `picks one minimizing an upper bound on its loss`

12     $t^* \leftarrow \arg\min_{t' \in V_i} \left( d^c(i, x_{t'}) + c \cdot r_{t'} + \min_{j \in \widehat{C}_{t'}} (d^m(i, j) - d^c(j, x_{t'})) \right)$;

      `// And switches to it if that improves the upper bound on its loss`

13     **if** $d^c(i, x_{t^*}) + c \cdot r_{t^*} + \min_{j \in \widehat{C}_{t^*}} (d^m(i, j) - d^c(j, x_{t^*})) < c \cdot r_t$ **then**

14       $C_t \leftarrow C_t \setminus \{i\}$;

15       $C_{t^*} \leftarrow C_{t^*} \cup \{i\}$;

16   **return** $\mathcal{X} = \{(C_1, x_1), \ldots, (C_k, x_k)\}$

---

The following two are the most significant and the most intricate results of our work.

**Theorem 2.** *For the dual metric loss, there always exists a clustering in the 3-core.*

**Theorem 3.** *For the dual metric loss, a clustering in the $(3 + 2\sqrt{3})$-core can be computed in polynomial time.*

To understand Algorithm 1, first we need to introduce the *Most Cohesive Cluster* problem, defined by Caragiannis et al. [2] for non-centroid clustering. Adapted to our semi-centroid clustering setting, it simply requires finding the cluster $(C, x)$ with $|C| \geqslant n/k$ that minimizes the maximum loss incurred by any agent in $C$. This is defined and approximated for arbitrary loss functions as follows.

**Definition 3** ($\alpha$-MCC). For $\alpha \geqslant 1$, and given a set of agents $\mathcal{N}$, a set of centers $\mathcal{M}$, a loss function $\ell$, and a threshold $\theta$ (typically $n/k$), a cluster $(C \subseteq \mathcal{N}, x \in \mathcal{M})$ with $|C| \geqslant \min(|\mathcal{N}|, \theta)$ is an $\alpha$-approximate most cohesive cluster (in short, an $\alpha$-MCC cluster) if $\max_{i \in C} \ell_i(C, x) \leqslant \alpha \cdot \max_{i \in S} \ell_i(S, y)$ for every cluster $(S \subseteq \mathcal{N}, y \in \mathcal{M})$ with $|S| \geqslant \min(|\mathcal{N}|, \theta)$. When $\alpha = 1$, it is simply called a most cohesive cluster.

Caragiannis et al. [2] use this in place of the greedy capture subroutine, which finds the cluster $(C, x)$ with the smallest radius $\max_{i \in C} d(i, x)$, in order to obtain improved fairness guarantees. Theorems 2 and 3 are both obtained using Algorithm 1, which calls a subroutine that finds an $\alpha$-MCC cluster; Theorem 2 is obtained by an inefficient subroutine that finds an MCC cluster, while Theorem 3 is obtained by plugging in an efficient subroutine we design for computing a 4-MCC cluster.

**Algorithm description:** Algorithm 1 is split into two phases. Phase 1 works by iteratively finding an $\alpha$-MCC cluster with respect to the dual metric loss and adding it to a tentative solution. We denote the clustering found in Phase 1 by $\widehat{\mathcal{X}} = \{(\widehat{C}_1, x_1), \ldots, (\widehat{C}_k, x_k)\}$. For each cluster $(\widehat{C}_t, x_t) \in \widehat{\mathcal{X}}$, the algorithm stores the maximum loss of any agent in that cluster as $r_t = \max_{i \in \widehat{C}_t} \ell_i(\widehat{C}_t, x_t)$.

Phase 2 takes the clustering $\widehat{\mathcal{X}}$ as a starting point and allows each agent $i \in \mathcal{N}$ to switch from its $t$-th cluster to any $t'$-th cluster subject to two careful constraints:

- *Bound the loss of the agents in the new cluster*: We know the original loss of every $j \in \widehat{C}_{t'}$ was at most $r_{t'}$. If agent $i$ switches to this cluster, it can increase $j$'s loss to $d^c(j, x_{t'}) + d^m(j, i)$. We need this to be at most $c \cdot r_{t'}$ for some constant $c$ given as input to the algorithm.
- *Bound own loss*: We show that the expression in Line 12—$d^c(i, x_{t'}) + c \cdot r_{t'} + \min_{j \in \widehat{C}_{t'}} (d^m(i, j) - d^c(j, x_{t'}))$—serves as an upper bound on the loss of agent $i$ when switching to the $t'$-th cluster, regardless of which other agents switch to the $t'$-th cluster during Phase 2. This ensures that agent $i$'s decision to switch (or not) is independent of the decisions made by the other agents in Phase 2, and the loss of agent $i$, who was originally in $\widehat{C}_t$, remains bounded by $c \cdot r_t$ throughout Phase 2, regardless of Phase 2 decisions. Agent $i$ makes a switch only if the new loss upper bound is lower than $c \cdot r_t$.

The key to establishing an approximate core guarantee lies in this selective switching administered in Phase 2. Intuitively, this can be seen as striking a middle ground between the centroid greedy capture, which allows an agent to switch to any other cluster because it cannot increase the *centroid loss* of the agents in that cluster, and the non-centroid greedy capture, which forbids such switches completely lest it might harm the agents in the cluster being switched to. The growth factor $c$ carefully controls the balance, and optimally setting this factor allows us to derive the following bound. Its proof, our most technically deep result, is presented in Appendix D.1.

**Lemma 1.** *Given a subroutine for computing an $\alpha$-approximate most cohesive cluster, Algorithm 1, with the growth factor set to $c = \frac{3\alpha + \sqrt{\alpha(\alpha + 8)}}{4\alpha}$, finds a clustering in the $\frac{1}{2}(\alpha + \sqrt{\alpha(\alpha + 8)} + 2)$-core with respect to the dual metric loss.*

From Lemma 1, the proof of Theorem 2 follows easily.

While Theorem 2 ensures the existence of a clustering in the 3-core, its computation relies on iteratively finding the most cohesive cluster, which is already NP-hard in the non-centroid paradigm [2]. Luckily, Lemma 1 yields a fairness guarantee even if we can find only an approximate most cohesive cluster during Phase 1 of Algorithm 1.

Next, we show how to compute a 4-approximate most cohesive cluster in polynomial time. The algorithm, presented as Algorithm 4 in Appendix D.1, works as follows. For each feasible cluster center $y \in \mathcal{M}$, it runs the non-centroid greedy capture with a crafted distance metric $d_y$, given by $d_y(i, j) = d^m(i, j) + d^c(i, y) + d^c(j, y)$ for all $i, j \in \mathcal{N}$, until it produces the first cluster $C_y$. Then, it returns $(C_{y^*}, y^*)$ with the lowest maximum loss, i.e., with $y^* \in \arg\min_{y \in \mathcal{M}} \max_{i \in C_y} \ell_i(C_y, y)$.

**Lemma 2.** *For the dual metric loss, Algorithm 4 computes a 4-approximate most cohesive cluster in polynomial time.*

Plugging this into Lemma 1 gives us Theorem 3, a constant approximation to the core in polynomial time.

**Lower bounds:** One may wonder how good these approximations are. As stated previously, semi-centroid clustering is more challenging than both centroid and non-centroid clustering. Specifically, one can trivially import a lower bound from either paradigm by setting $d^m = 0$ or $d^c = 0$ (where a distance metric being zero means all pairwise distances are zero), which makes the dual metric loss coincide with the centroid or non-centroid loss, respectively. For the non-centroid loss, Caragiannis et al. [2] do not offer any lower bounds; that is, the existence of an exact core clustering with respect to the (maximum distance) non-centroid loss remains an open question. However, Chen et al. [1] establish a lower bound of 2-core with respect to the centroid loss, which we can import. Somewhat surprisingly, we are unable to derive a lower bound better than 2 despite reasonable effort.

### 4.2 Weighted Single Metric Loss

In this section, we focus on improving the core approximation guarantees for the restricted case of the weighted single metric loss. Recall that this is parametrized by a single metric $d$ and a parameter $\lambda \in [0, 1]$ such that, for any $i \in \mathcal{N}$ and cluster $(C, x)$, we have $\ell_i(C, x) = \lambda \cdot \max_{j \in C} d(i, j) + (1 - \lambda) \cdot d(i, x)$. We devise two algorithms that offer improvements.

The first algorithm takes non-centroid greedy capture (Algorithm 3), which grows balls around agents, and adapts it to the semi-centroid world by assigning each cluster to the cluster center nearest to the agent whose ball captured the cluster. The resulting efficient algorithm is presented as Algorithm 5 in Appendix D.2, and achieves the following guarantee.

**Lemma 3.** *For the weighted single metric loss with parameter $\lambda \in (0, 1]$, Algorithm 5 finds a clustering in the $\frac{2}{\lambda}$-core in polynomial time.*

The second algorithm, presented formally as Algorithm 6 in Appendix D.2, is similar to Algorithm 1: in Phase 1, agents are partitioned into an initial clustering, and in Phase 2, agents are allowed to switch to a different cluster that reduces their loss and does not significantly increase the loss of the agents in that cluster. However, there are two key differences. First, Algorithm 6 uses the centroid greedy capture subroutine instead of the (approximate) most cohesive cluster subroutine used by Algorithm 1 to iteratively find clusters in Phase 1. Second, due to having a single metric, the switching condition is much simpler and leads to better bounds for some values of $\lambda$. A careful analysis yields the following guarantee.

**Lemma 4.** *For the weighted single metric loss with parameter $\lambda \in [0, 1)$, Algorithm 6 finds a clustering in the $\frac{\sqrt{2\lambda - 11\lambda^2 + 13} + 3 - \lambda}{2 - 2\lambda}$-core in polynomial time.*

Combining the bounds from Lemmas 3 and 4, together with the existential 3-core (Theorem 2) and the polynomial-time $(3 + 2\sqrt{3})$-core (Theorem 3) guarantees that we can import from dual metric loss, yields the following existential and efficiently attainable bounds.

**Theorem 4.** *For the weighted single metric loss with parameter $\lambda \in [0, 1]$, a clustering in the $\min\{\frac{2}{\lambda}, 3\}$-core always exists, and a clustering in the $\min\{\frac{2}{\lambda}, \frac{\sqrt{2\lambda - 11\lambda^2 + 13} + 3 - \lambda}{2 - 2\lambda}\}$-core can be found in polynomial time.*

**Lower bounds:** Similarly, we also derive two lower bounds on the core approximation under the weighted single metric loss. The first bound interpolates between $(\sqrt{5} + 1)/2$ at $\lambda = 0$ (centroid) and 1 at $\lambda = 1$ (non-centroid). Since Chen et al. [1] prove a lower bound of 2 for $\lambda = 0$ (centroid), which is better than $(\sqrt{5} + 1)/2$, adapting their counterexample to semi-centroid clustering yields a better lower bound when $\lambda$ is close to 0.

**Theorem 5.** *For the weighted single metric loss with parameter $\lambda \in [0, 1]$, no clustering algorithm can achieve a core approximation better than $\max\{\frac{\sqrt{\lambda^2 - 2\lambda + 5} - \lambda + 1}{2}, \frac{2(1 - \lambda)}{2\lambda + 1}\}$.*

## 5 Fully Justified Representation

Let us turn our attention to the weaker criterion of fully justified representation (FJR), a relaxation of the core originally introduced by Peters et al. [24] in the context of participatory budgeting. While we can already achieve a (small) constant approximation to the core even under the dual metric loss (Theorems 2 and 3), we show that this relaxation offers several advantages.

Caragiannis et al. [2] study FJR for non-centroid clustering, and show that iteratively finding an $\alpha$-approximate most cohesive cluster produces an $\alpha$-FJR clustering, even under *arbitrary loss functions*. The same result, with an almost identical proof, holds in our more general semi-centroid clustering setup. For completeness, in Appendix E, we formally present the algorithm as Algorithm 7 and provide a complete proof. This gives us the existence of a 1-FJR clustering under arbitrary loss functions, and polynomial-time computation of a 4-FJR clustering under the dual metric loss when plugging in our polynomial-time algorithm for computing a 4-MCC cluster (Lemma 2).

**Theorem 6.** *For arbitrary losses and $\alpha \geqslant 1$, Algorithm 7, which iteratively finds an $\alpha$-approximate most cohesive cluster, returns an $\alpha$-FJR clustering. Hence, a 1-FJR clustering exists for arbitrary losses and, due to Lemma 2, a 4-FJR clustering can be computed in polynomial time for the dual metric loss.*

In Section 3, we noticed that for a clustering, being in the $\alpha$-core with respect to both centroid and non-centroid losses simultaneously is (trivially) a necessary condition for it to be in the $\alpha$-core with respect to the weighted single metric loss for all $\lambda \in [0, 1]$.

For FJR, this is clearly still the case, and unfortunately, we can show that a simultaneous FJR approximation with respect to centroid and non-centroid losses induced by two different metrics is still infeasible.

**Theorem 7.** *For any $\alpha \geqslant 1$ and $\beta \geqslant 1$, there exists an instance in which no clustering is both $\alpha$-FJR with respect to the centroid loss and $\beta$-FJR with respect to the non-centroid loss, when the two loss functions are allowed to use different metrics.*

When both losses are induced by a common metric $d$, a surprisingly simple algorithm turns out to achieve a constant approximation to FJR with respect to both centroid and non-centroid losses. This is in contrast to the core, for which no simultaneous finite approximation is feasible, even for a single metric (Theorem 1).

The algorithm is Algorithm 5, which runs non-centroid greedy capture with metric $d$ followed by a greedy centroid selection step. We presented it in Section 4.2 and proved that it yields $2/\lambda$-core for all $\lambda \in [0, 1]$. Note that this achieves 2-core (and hence, 2-FJR) with respect to the non-centroid loss ($\lambda = 1$). However, its approximation to the core becomes unbounded as $\lambda$ approaches 0 (the centroid loss), which is inevitable given Theorem 1. However, we show that its approximation to the weaker criterion of FJR still remains bounded by 5.

**Theorem 8.** *Algorithm 5, given metric $d$, finds, in polynomial time, a clustering that is 5-FJR with respect to the centroid loss and 2-FJR with respect to the non-centroid loss, both defined using $d$.*

## 6   Discussion

Our work leaves open a number of immediate technical questions, which can be gleaned from the gaps between upper and lower bounds in Table 1. Future work can also investigate generalizations such as non-additive combinations of centroid and non-centroid losses and several exciting applications.

One particularly important special case of this model that we leave largely unexplored is when $\mathcal{M} = \mathcal{N}$. As mentioned previously, this restriction applied to centroid clustering is what Ebadian and Micha [17] use to model the problem of sortition (randomly selecting a committee of agents that represents the whole population), and can also be used to model many problems where one wishes to group a set of $n$ datapoints into $k$ clusters, and return a single point to serve as a representative point for that cluster. This can have numerous applications such as coreset selection [25], scalable LLM fine-tuning [26], feature selection [27], and dimensionality reduction [28]. However, our semi-centroid algorithms offer no guarantees that the centroid it selects for a cluster will be a point that is in that cluster, which may be desirable in all these cases. Exploring algorithms with this additional constraint would be an interesting future direction.

Some of these applications may be large scale, for which our polynomial-time algorithms may not suffice and near-linear time algorithms may need to be devised. Investigating proportional fairness guarantees attainable via such ultra-fast algorithms remains an uncharted territory.

## Acknowledgments and Disclosure of Funding

This work was supported by an NSERC Discovery Grant and an NSERC-CSE Research Communities Grant. Researchers funded through the NSERC-CSE Research Communities Grants do not represent the Communications Security Establishment Canada or the Government of Canada. Any research, opinions or positions they produce as part of this initiative do not represent the official views of the Government of Canada.

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

# Appendix

## A  Missing Proofs from Section 2

**Proposition 1.** *For every $\alpha \geqslant 1$, every clustering in the $\alpha$-core is also $\alpha$-FJR.*

*Proof.* For contradiction, assume this is false. For some instance, there is a clustering $\mathcal{X} = \{(C_1, x_1), \ldots, (C_k, x_k)\}$ and an $\alpha \geqslant 1$ such that $\mathcal{X}$ is in the $\alpha$-core, but is not $\alpha$-FJR. By the FJR definition, this means that there is a group of agents $S \subseteq N$ with $|S| \geqslant n/k$ and a center $y \in \mathcal{M}$ such that $\alpha \cdot \ell_i(S, y) < \min_{j \in S} \ell_j(\mathcal{X})$ for all $i \in S$.

It is simple to see that for each $i \in S$, $\alpha \cdot \ell_i(S, y) < \min_{j \in S} \ell_j(\mathcal{X})$ implies that $\alpha \cdot \ell_i(S, y) < \ell_i(\mathcal{X})$, meaning that $(S, y)$ induces an $\alpha$-core violation as well, yielding a contradiction. $\square$

## B  Greedy Capture Algorithms

---

**Algorithm 2:** Centroid Greedy Capture [1]

**Input:** $\mathcal{N}, \mathcal{M}, d, k$;
**Output:** $\mathcal{X} = \{(C_1, x_1), \ldots, (C_k, x_k)\}$.
Initialize: $\mathcal{N}' \leftarrow \mathcal{N}, \mathcal{X} \leftarrow \emptyset, \delta \leftarrow 0$
`// Uncaptured agents remain`
**while** $\mathcal{N}' \neq \emptyset$ **do**
    Increase $\delta$ smoothly
    `// Ball opens at a center`
    **if** $\exists x \in \mathcal{M}, C \subseteq \mathcal{N}'$ *s.t.* $|C| \geqslant n/k$
    *and* $d(i, x) \leqslant \delta$ *for all* $i \in C$ **then**
        $\mathcal{X} \leftarrow \mathcal{X} \cup \{(C, x)\}$
        $\mathcal{N}' \leftarrow \mathcal{N}' \setminus C$
    `// Open balls keep growing`
    **if** $\exists (C, x) \in \mathcal{X}, i \in \mathcal{N}'$ *s.t.*
    $d(i, x) \leqslant \delta$ **then**
        $(C, x) \leftarrow (C \cup \{i\}, x)$
        $\mathcal{N}' \leftarrow \mathcal{N}' \setminus \{i\}$

`// Agents switch clusters`
$P \leftarrow \{x \in \mathcal{M} : (C, x) \in \mathcal{X}$ for some $C\}$
$\forall i \in \mathcal{N}, x_i \in \arg\min_{x \in P} d(i, x)$
$\forall x \in P, \widehat{C}_x \leftarrow \{i \in \mathcal{N} : x_i = x\}$
$\mathcal{X} \leftarrow \{(\widehat{C}_x, x) : x \in P\}$
**return** $\mathcal{X}$

---

**Algorithm 3:** Non-Centroid Greedy Capture [2]

**Input:** $\mathcal{N}, d, k$;
**Output:** $\mathcal{X} = \{C_1, \ldots, C_k\}$.
Initialize: $\mathcal{N}' \leftarrow \mathcal{N}, \mathcal{X} \leftarrow \emptyset, \delta \leftarrow 0$
`// Uncaptured agents remain`
**while** $\mathcal{N}' \neq \emptyset$ **do**
    Increase $\delta$ smoothly
    `// Ball opens at an agent`
    **if** $\exists i \in \mathcal{N}', C \subseteq \mathcal{N}'$ *s.t.*
    $|C| \geqslant \min(|\mathcal{N}'|, n/k)$ *and* $d(i, i') \leqslant \delta$ *for*
    *all* $i' \in C$ **then**
        $\mathcal{X} \leftarrow \mathcal{X} \cup \{C\}$
        $\mathcal{N}' \leftarrow \mathcal{N}' \setminus C$
    `// Open balls don't grow`

`// Agents don't switch clusters`
**return** $\mathcal{X}$

---

In the three dimensions discussed in Section 3, which induce eight variants of greedy capture, centroid greedy capture (Algorithm 2) makes the following choices:

(1) The balls grow around the centroids.

(2) After a ball "captures" a group of $n/k$ points and opens, it continues to grow and capture any uncaptured points as soon as it contains them.

(3) After all agents have been captured, each agent gets an opportunity to switch to the cluster containing the centroid that is closest to it.

We can show that the $(1 + \sqrt{2})$-core guarantee of this algorithm with respect to the centroid loss is retained even when disabling one of the features (2) and (3).

**Theorem 9.** *Consider a variant of greedy capture in which balls grow around the agents, and open balls keep growing and/or agents are allowed to switch clusters at the end of the algorithm (at least*

*one of the two holds). Then, it always returns a clustering in the $(1 + \sqrt{2})$-core with respect to the centroid loss.*

*Proof.* For contradiction assume this is false, let $\mathcal{X} = \{(C_1, x_1), \ldots, (C_k, x_k)\}$ be the clustering found by such an algorithm, and suppose that there is a deviating cluster $(S, y)$ where $|S| \geqslant \lceil \frac{n}{k} \rceil$ and $(1 + \sqrt{2}) \cdot d(i, y) < d(i, x_{\mathcal{X}(i)})$ for every $i \in S$.

Let $i = \arg\min_{i' \in S} \mathcal{X}(i')$ be the first agent in $S$ that was captured by the algorithm, and let $t = \mathcal{X}(i)$. We know that $x_t \neq y$, or $i$ would clearly not be a part of the deviating cluster $S$. Let $j = \arg\max_{j' \in S} d(j', y)$ be the agent in $S$ that is farthest from $y$, and let $\mathcal{X}(j) = t'$.

First, note that we must have $d(j, y) \geqslant d(i, x_t)$. If this were false, and $d(j, y) < d(i, x_t)$, then by the way that greedy capture functions, the cluster $(S, y)$ would be captured prior to $i$ being captured by $x_t$, which would give a contradiction.

With the above inequality, we can show that $t \neq t'$. If this were not the case, and $t = t'$, then we could say the following:

$$
\begin{aligned}
(1 + \sqrt{2}) \cdot d(j, y) < d(j, x_{t'}) &= d(j, x_t) \\
&\leqslant d(i, x_t) + d(i, y) + d(j, y) \text{ //Triangle inequality} \\
&< d(i, x_t) + \frac{1}{1 + \sqrt{2}} \cdot d(i, x_t) + d(j, y) \\
&\leqslant \left( \frac{1}{1 + \sqrt{2}} + 2 \right) \cdot d(j, y).
\end{aligned}
$$

Hence, $1 + \sqrt{2} < \frac{1}{1+\sqrt{2}} + 2 = 1 + \sqrt{2}$, which is a contradiction.

We can also show that $d(j, x_t) \geqslant d(j, x_{t'})$. We can do this by considering two cases.

*Case 1: Open balls keep growing.* If open balls keep growing in the greedy capture variant under consideration, and if $d(j, x_t) < d(j, x_{t'})$, then the fact that $x_t$ captures $i$ prior to $x_{t'}$ capturing $j$ implies that at the point in the algorithm when $x_{t'}$ captures $j$, $x_t$ will have already captured $i$, and $j$ will have already been in the radius of $x_t$'s ball at a previous value of $\delta$. This would mean that $x_t$ must have initially captured $j$. But this would contradict $\mathcal{X}(j) = t'$, since regardless of if agents are allowed to switch clusters in the last step of the algorithm, $j$ could not end up in the $t'$-th cluster at the end of the algorithm.

*Case 2: Agents can switch clusters.* If agents can switch clusters at the end of the greedy capture variant under consideration, and if $d(j, x_t) < d(j, x_{t'})$, then regardless of whether open balls continue to grow, $j$ would always want to switch to $x_t$ over $x_{t'}$, contradicting the fact that $\mathcal{X}(j) = t'$.

With this inequality, we can say the following:

$$
\begin{aligned}
(1 + \sqrt{2}) \cdot d(j, y) &< d(j, x_{t'}) \\
&\leqslant d(j, x_t) \\
&\leqslant d(i, x_t) + d(i, y) + d(y, j) \\
&< d(i, x_t) + \frac{1}{1 + \sqrt{2}} \cdot d(i, x_t) + d(j, y) \\
&\leqslant \left( \frac{1}{1 + \sqrt{2}} + 2 \right) \cdot d(j, y),
\end{aligned}
$$

Again yielding $1 + \sqrt{2} < \frac{1}{1+\sqrt{2}} + 2 = 1 + \sqrt{2}$, which is a contradiction. $\square$

## C   Missing Proofs from Section 3

**Theorem 1.** *For every $\alpha \geqslant 1$ and $\beta \geqslant 1$, there exists a semi-centroid clustering instance in which no clustering is simultaneously in the $\alpha$-core with respect to the centroid loss and in the $\beta$-core with respect to the non-centroid loss.*

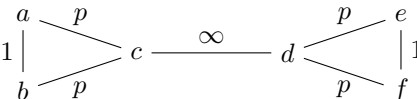

Figure 1: A semi-centroid clustering instance in which no clustering yields a finite approximation to the core with respect to both the centroid and non-centroid losses simultaneously.

*Proof.* Fix any $\alpha \geqslant 1$, $\beta \geqslant 1$, and $p > \max\{\alpha, \beta\}$. Consider the instance in Figure 1. Here $\mathcal{N} = \{a, b, c, d, e, f\}$, $\mathcal{M} = \mathcal{N}$, $k = 3$, and let $d^m = d^c = d$, where $d$ is defined by the distances shown in the diagram,[2] with each distance that is not shown assumed to be the maximum possible allowed under the triangle inequality. Since $n/k = 2$, a deviating coalition must contain 2 or more agents.

For contradiction, assume that there exists a clustering in this instance that is simultaneously in the $\alpha$-core in the centroid paradigm and in the $\beta$-core in the non-centroid paradigm.

*Non-centroid $\beta$-core:* We claim that $\{a, b\}$ must be one of the clusters. If not, then either $a$ and $b$ are each in a cluster with a different agent, or at least one of them is a singleton cluster. In the former case, $\{a, b\}$ can deviate and improve the non-centroid loss of each member by a factor of $p > \beta$, which is a contradiction. In the latter case, since $k = 3$, by the pigeonhole principle, there must exist a cluster $C$ with $|C| \geqslant 3$ (which means $C \cap \{d, e, f\} \neq \emptyset$). If $C \cap \{a, b, c\} \neq \emptyset$, then the larger of $C \cap \{a, b, c\}$ and $C \cap \{d, e, f\}$, which must contain at least two agents, can deviate and each of its members would improve by an infinite factor, again a contradiction. The only remaining possibility is that $C = \{d, e, f\}$, in which case $\{e, f\}$ can deviate and improve each member by a factor of $p > \beta$, a contradiction.

Hence, $\{a, b\}$ must be one of the clusters. By symmetry, $\{e, f\}$ must also be a cluster, so the remaining cluster must be $\{c, d\}$. Thus, the only clustering that can be in the $\beta$-core in the non-centroid paradigm is $\{(\{a, b\}, x_1), (\{c, d\}, x_2), (\{e, f\}, x_3)\}$ for some $x_1, x_2, x_3 \in \mathcal{N}$.

*Centroid $\alpha$-core:* If $x_1 \notin \{a, b\}$, then $\{a, b\}$ can deviate with either of $a$ or $b$ as the cluster center and improve the centroid loss of each by a factor at least $p > \alpha$, a contradiction. Without loss of generality, assume $x_1 = a$. Similarly, assume without loss of generality that $x_3 = e$. Now, either $x_2 \in \{a, b, c\}$ or $x_2 \in \{d, e, f\}$. In the former case, $\{d, f\}$ can deviate with a center at $f$, and in the latter case, $\{b, c\}$ can deviate with a center at $b$, in each case improving each member by a factor at least $p > \alpha$, which is a contradiction.

This concludes the proof. $\qquad\square$

# D    Missing Proofs from Section 4

## D.1    Dual Metric Loss

Here, we present the missing proof of our central result, Lemma 1, which establishes an approximate core guarantee for Algorithm 1 with access to an oracle for an $\alpha$-approximate maximum cohesive cluster. In the main body, we use it to prove our existential and polynomial-time attainable approximate core guarantees for the dual metric loss (Theorems 2 and 3). Let us recall the statement.

**Lemma 1.** *Given a subroutine for computing an $\alpha$-approximate most cohesive cluster, Algorithm 1, with the growth factor set to $c = \frac{3\alpha + \sqrt{\alpha(\alpha + 8)}}{4\alpha}$, finds a clustering in the $\frac{1}{2}(\alpha + \sqrt{\alpha(\alpha + 8)} + 2)$-core with respect to the dual metric loss.*

We begin by establishing several useful properties of Algorithm 1. These properties bound the loss incurred by the agents when they (or other agents) switch (or choose not to switch) clusters during Phase 2 of Algorithm 1.

Recall that Phase 1 produces a tentative clustering $\widehat{\mathcal{X}}$, with $r_t = \max_{i \in \widehat{C}_t} \ell_i(\widehat{C}_t, x_t)$ being the maximum loss in the $t$-th tentative cluster, $(\widehat{C}_t, x_t)$. Then, in Phase 2, each agent $i \in \widehat{C}_t$ is given the

---

[2]The infinite distances can be replaced by very large distances without affecting the proof, but we keep them infinite for simplicity.

opportunity to switch to the $t'$-th cluster for every $t' \in V_i$. For each such cluster, the agent evaluates the expression

$$\Phi(i, t') \triangleq d^c(i, x_{t'}) + c \cdot r_{t'} + \min_{j \in \widehat{C}_{t'}} (d^m(i, j) - d^c(j, x_{t'})),$$

picks its minimizer $t^* \in \arg\min_{t' \in V_i} \Phi(i, t')$, and switches to it if $\Phi(i, t') < c \cdot r_t$. The following claims shed light into the reasoning behind this specific step.

The first lemma establishes that if an agent is part of the $t$-th cluster at the end of both Phase 1 and Phase 2 (i.e., it does not switch clusters during Phase 2), then its loss, which is at most $r_t$ at the end of Phase 1, remains bounded by $c \cdot r_t$ at the end of Phase 2. The increase may come from other agents possibly switching into the $t$-th cluster during Phase 2.

**Lemma 5.** *For each $i \in \mathcal{N}$, if $\widehat{\mathcal{X}}(i) = \mathcal{X}(i) = t$, then $\ell_i(\mathcal{X}) = \ell_i(C_t, x_t) \leqslant c \cdot r_t$.*

*Proof.* Our goal is to show that $d^c(i, x_t) + d^m(i, j) \leqslant c \cdot r_t$ for all $j \in C_t$. Consider any $j \in C_t$.

*Case 1: $j \in C_t \cap \widehat{C}_t$.* By the definition of $r_t$, we must have $d^c(i, x_t) + d^m(i, j) \leqslant r_t \leqslant c \cdot r_t$, where the last inequality holds because $c \geqslant 1$. In words, such an agent does not increase the loss of agent $i$ even slightly.

*Case 2: $j \in C_t \setminus \widehat{C}_t$.* Because agent $j$ was allowed to switch to the $t$-th cluster, we must have $t \in V_j$. Then, Line 11 in Algorithm 1 ensures that $d^c(i, x_t) + d^m(i, j) \leqslant c \cdot r_t$; this is the condition that forbids an agent from joining a cluster if it would significantly hurt a "core member" of that cluster.

This concludes the proof. □

The next lemma shows that if agent $i$ switches to the $t^*$-th cluster during Phase 2, its final loss $\ell_i(\mathcal{X})$ would be bounded by $\Phi(i, t^*)$.

**Lemma 6.** *For each $i \in \mathcal{N}$, if $\mathcal{X}(i) = t^* \neq \widehat{\mathcal{X}}(i)$, then*

$$\ell_i(\mathcal{X}) = \ell_i(C_{t^*}, x_{t^*}) \leqslant \Phi(i, t^*) = d^c(i, x_{t^*}) + c \cdot r_{t^*} + \min_{j \in \widehat{C}_{t^*}} (d^m(i, j) - d^c(j, x_{t^*})).$$

*Proof.* Let $i' = \arg\max_{j \in C_{t^*}} d^m(i, j)$ be the agent in $C_{t^*}$ that is the farthest from $i$. Then, we have $\ell_i(C_{t^*}, x_{t^*}) = d^c(i, x_{t^*}) + d^m(i, i')$. We want to show that this is at most $\Phi(i, t^*)$. Let $j^* = \arg\min_{j \in \widehat{C}_{t^*}} (d^m(i, j) - d^c(j, x_{t^*}))$, so $\Phi(i, t^*) = d^c(i, x_{t^*}) + c \cdot r_{t^*} + d^m(i, j^*) - d^c(j^*, x_{t^*})$. Let us simplify our goal:

$$d^c(i, x_{t^*}) + d^m(i, i') \leqslant d^c(i, x_{t^*}) + c \cdot r_{t^*} + d^m(i, j^*) - d^c(j^*, x_{t^*})$$
$$\Leftrightarrow d^m(i, i') \leqslant c \cdot r_{t^*} + d^m(i, j^*) - d^c(j^*, x_{t^*}). \tag{1}$$

*Case 1: $i' \in \widehat{C}_{t^*}$.* Then, by the fact that $j^* \in \widehat{C}_{t^*}$ is also true, we must have that $d^c(j^*, x_{t^*}) + d^m(j^*, i') \leqslant \ell_{j^*}(\widehat{C}_{t^*}, x_{t^*}) \leqslant r_{t^*}$, which yields $d^m(j^*, i') \leqslant r_{t^*} - d^c(j^*, x_{t^*})$.

*Case 2: $i' \notin \widehat{C}_{t^*}$.* Then, since agent $i'$ switched to the $t^*$-th cluster during Phase 2, we must have $t^* \in V_{i'}$. Line 11 of Algorithm 1 and the fact that $j^* \in \widehat{C}_{t^*}$ ensure that $d^c(j^*, x_{t^*}) + d^m(j^*, i') \leqslant c \cdot r_{t^*}$, i.e., $d^m(j^*, i') \leqslant c \cdot r_{t^*} - d^c(j^*, x_{t^*})$.

Since $c \geqslant 1$, we get $d^m(j^*, i') \leqslant c \cdot r_{t^*} - d^c(j^*, x_{t^*})$ in both cases. Using the triangle inequality, we get

$$d^m(i, i') \leqslant d^m(i, j^*) + d^m(j^*, i') \leqslant d^m(i, j^*) + c \cdot r_{t^*} - d^c(j^*, x_{t^*}),$$

which is the desired Equation (1), concluding the proof. □

Here is what we know now. During Phase 2, if agent $i$ remains in its original cluster $\widehat{\mathcal{X}}(i) = t$, its loss remains upper bounded by $c \cdot r_t$ (Lemma 5), and if it switches to a cluster $t' \in V_i$, its loss remains upper bounded by $\Phi(i, t')$. This explains why agent $i$ switches to the cluster $t^* \in \arg\min_{t' \in V_i} \Phi(i, t')$, only if $\Phi(i, t^*) < c \cdot r_t$: the agent is simply picking the cluster that offers the smallest upper bound on its eventual loss. This is captured by the next lemma.

**Lemma 7.** *For each $i \in \mathcal{N}$ and $t' \in V_i$, we have $\ell_i(\mathcal{X}) \leqslant \Phi(i, t') = d^c(i, x_{t'}) + c \cdot r_{t'} + \min_{j \in \widehat{C}_{t'}} (d^m(i, j) - d^c(j, x_{t'}))$.*

*Proof.* Let $t = \widehat{\mathcal{X}}(i)$ and $t^* \in \arg\min_{t' \in V_i} \Phi(i, t')$. If agent $i$ switches to the $t^*$-th cluster during Phase 2 of Algorithm 1, then the claim follows from the fact that $\Phi(i, t^*) \leqslant \Phi(i, t')$ for all $t' \in V_i$. If agent $i$ does not make a switch during Phase 2 and remains in the $t$-th cluster, this must be because $c \cdot r_t \leqslant \Phi(i, t^*)$ due to Line 13 of Algorithm 1. The claim again follows from $\Phi(i, t^*) \leqslant \Phi(i, t')$ for all $t' \in V_i$. $\qquad\square$

Similarly to the proof of Lemma 7, we can see that if agent $i$ was in the $t$-th cluster at the end of Phase 1, then $c \cdot r_t$ also remains an upper bound on its eventual loss, regardless of whether the agent stays in the $t$-th cluster or switches to a $t^*$-th cluster during Phase 2. This is because the agent only switches to the $t^*$-th cluster if it provides a smaller upper bound on the loss.

**Lemma 8.** *For each $i \in \mathcal{N}$, if $\widehat{\mathcal{X}}(i) = t$, then $\ell_i(\mathcal{X}) \leqslant c \cdot r_t$.*

*Proof.* If $\mathcal{X}(i) = t$ (i.e., the agent stays in the $t$-th cluster during Phase 2), then the claim reduces to Lemma 5. If $\mathcal{X}(i) = t^* \neq t$, then by Lemma 6, we have $\ell_i(\mathcal{X}) \leqslant \Phi(i, t^*)$. However, due to Line 13, agent $i$ only switches to the $t^*$-th cluster if $\Phi(i, t^*) < c \cdot r_t$, which implies $\ell_i(\mathcal{X}) \leqslant c \cdot r_t$, as needed. $\qquad\square$

Together, Lemmas 7 and 8 give us two different ways to upper bound the loss of agent $i$, in terms of the $t$-th cluster that agent $i$ was part of at the end of Phase 1, and in terms of every $t'$-th cluster in $V_i$ that agent $i$ had the opportunity to switch to during Phase 2. With these bounds, we are ready to prove the central lemma.

*Proof.* Proof of Lemma 1

Fix an $\alpha \geqslant 1$ and let $\beta = \frac{1}{2}(\alpha + \sqrt{\alpha(\alpha + 8)} + 2)$.

For contradiction, assume that in some clustering $\mathcal{X} = \{(C_1, x_1), \ldots, (C_k, x_k)\}$ returned by the algorithm, there is some deviating cluster $(S, y)$ such that for all $i \in S, \ell_i(S, y) < \frac{1}{\beta}\ell_i(\mathcal{X})$. Let $i = \arg\min_{i \in S} \widehat{\mathcal{X}}(i)$ be the agent from this deviating cluster that was captured first during phase 1 of Algorithm 1. Let $\widehat{\mathcal{X}}(i) = t$. Let $j = \arg\max_{j \in S} \ell_j(S, y)$ be the agent from $S$ that has the highest loss among all agents in the deviating cluster.

First note that it must be the case that $\ell_j(S, y) \geqslant \frac{1}{\alpha}r_t$. If this were not the case, it would contradict the fact that $(\widehat{C}_t, x_t)$ was a $\alpha$-approximation of the most cohesive cluster at that step of the algorithm.

Next, we can show that $\widehat{\mathcal{X}}(j) \neq t$, and that $j$ did not have the opportunity to switch into the $t$-th cluster during the second phase of the algorithm.

Showing that $\widehat{\mathcal{X}}(j) \neq t$ is easy. If $\widehat{\mathcal{X}}(j) = t$ were true, then by Lemma 8, we must have that $\ell_j(\mathcal{X}) \leqslant c \cdot r_t$. Since $j$ wants to deviate to $(S, y)$, we also must have that $\ell_j(S, y) < \frac{1}{\beta}\ell_j(\mathcal{X}) \leqslant \frac{c}{\beta}r_t$. By plugging in the correct values of $\beta$ and $c$, it can be seen that $\frac{c}{\beta} < \frac{1}{\alpha}$ whenever $\alpha \geqslant 1$, so this would contradict that $\ell_j(S, y) \geqslant \frac{1}{\alpha}r_t$.

With a bit more work, we can also show that $j$ did not have the opportunity to switch to the $t$-th cluster during the second phase of the algorithm. If $j$ did have this opportunity, then by Lemma 7 we must have that $\ell_j(\mathcal{X}) \leqslant d^c(j, x_t) + c \cdot r_t + \min_{i' \in \widehat{C}_t}\{d^m(i', j) - d^c(i', x_t)\}$. Since $i \in \widehat{C}_t$, this also gives us $\ell_j(\mathcal{X}) \leqslant d^c(j, x_t) + c \cdot r_t + d^m(i, j) - d^c(i, x_t)$.

By the triangle inequality, we have that $d^c(j, x_t) \leqslant d^c(j, y) + d^c(i, y) + d^c(i, x_t)$. Plugging this into our bound on $j$'s loss gives us:

$$\ell_j(\mathcal{X}) \leqslant c \cdot r_t + d^m(i, j) + d^c(j, y) + d^c(i, y)$$

Notice that since $i$ and $j$ are both in $S$, $d^m(i, j) + d^c(j, y) \leqslant \ell_j(S, y)$. Also, we must have that $d^c(i, y) \leqslant \ell_i(S, y) < \frac{1}{\beta}\ell_i(\mathcal{X}) \leqslant \frac{c}{\beta}r_t$, where the last inequality follows from Lemma 8. Plugging these in to our inequality gives us:

$$\ell_j(\mathcal{X}) < \ell_j(S, y) + \left(c + \frac{c}{\beta}\right) \cdot r_t.$$

This implies that when $j$ deviates to $(S, y)$, it can only improve its loss by a factor of at most:

$$\frac{\ell_j(S, y) + \left(c + \frac{c}{\beta}\right) \cdot r_t}{\ell_j(S, y)}.$$

The fact that $\ell_j(S, y) \geqslant \frac{1}{\alpha} r_t$ allows us to upper-bound this as

$$\frac{\frac{1}{\alpha} \cdot r_t + \left(c + \frac{c}{\beta}\right) \cdot r_t}{\frac{1}{\alpha} \cdot r_t} = 1 + c \cdot \alpha \cdot \left(1 + \frac{1}{\beta}\right).$$

By substituting in the values of $c$ and $\beta$ in terms of $\alpha$, this comes out to be $\frac{1}{2}(\alpha + \sqrt{\alpha(\alpha + 8)} + 2)$, which gives a contradiction.

Thus, it must be the case that $j$ does not have the opportunity to switch to the $t$-th cluster during the second phase of the algorithm. Therefore, there must be some $i' \in \widehat{C}_t$ such that $d^c(i', x_t) + d^m(i', j) > c \cdot r_t$, which can be rearranged to get $d^m(i', j) > c \cdot r_t - d^c(i', x_t)$. As the final part of the proof, we can show that this also leads to a contradiction.

By the triangle inequality, we have that $d^m(i', j) \leqslant d^m(i', i) + d^m(i, j)$. This can be rearranged to get $d^m(i, j) \geqslant d^m(i', j) - d^m(i', i)$. Plugging $d^m(i', j) > c \cdot r_t - d^c(i', x_t)$ into this inequality gives us $d^m(i, j) > c \cdot r_t - (d^c(i', x_t) + d^m(i', i))$. Since $i$ and $i'$ are both in $\widehat{C}_t$, we have that $d^c(i', x_t) + d^m(i', i) \leqslant \ell_{i'}(\widehat{C}_t, x_t) \leqslant r_t$, which gives us $d^m(i, j) > (c - 1)r_t$. Since $i$ and $j$ are both in $S$, this tells us that $\ell_i(S, y) \geqslant d^m(i, j) > (c - 1)r_t$.

Finally, since $i \in \widehat{C}_t$, by Lemma 8 we know that $\ell_i(\mathcal{X}) \leqslant c \cdot r_t$. This means that when $i$ deviates to $(S, y)$, they are only improving their loss by a factor less than $\frac{c \cdot r_t}{(c-1)r_t} = \frac{1}{2}(\alpha + \sqrt{\alpha(\alpha + 8)} + 2)$, giving another contradiction and completing the proof. $\qquad \square$

From Lemma 1, the proof of Theorem 2 follows rather easily.

**Theorem 2.** *For the dual metric loss, there always exists a clustering in the 3-core.*

*Proof of Theorem 2.* Set $\alpha = 1$ (i.e., let Algorithm 1 iteratively compute a most cohesive cluster in Phase 1) in Lemma 1, which sets the growth factor to $c = 3/2$ and yields a 3-approximation to the core. $\qquad \square$

Next, we present Algorithm 4, which efficiently computes a 4-MCC cluster for the dual metric loss by running the non-centroid greedy capture algorithm on a carefully crafted distance metric, once for each feasible center.

---

**Algorithm 4:** 4-Approximate Most Cohesive Clustering Algorithm

---

**Input:** Set of agents $\mathcal{N}$, set of feasible centers $\mathcal{M}$, non-centroid metric $d^m$, centroid metric $d^c$, number of clusters $k$

**Output:** Cluster $(C, x)$

1 **for** $y \in \mathcal{M}$ **do**
2 $\quad \forall i, j \in \mathcal{N} : d_y(i, j) \leftarrow d^m(i, j) + d^c(i, y) + d^c(j, y)$;      // Craft the metric
3 $\quad C_y \leftarrow$ First cluster produced by Algorithm 3 with input $(\mathcal{N}, d_y, k)$;    // Run greedy capture
4 $\quad r_y \leftarrow \max_{i \in C_y} \ell_i(C_y, y)$;      // $\ell_i$ = dual metric loss
5 $y^* \leftarrow \arg\min_{y \in \mathcal{M}} r_y$;      // Best center
6 **return** $(C_{y^*}, y^*)$

---

The proof of its approximation guarantee follows.

**Lemma 2.** *For the dual metric loss, Algorithm 4 computes a 4-approximate most cohesive cluster in polynomial time.*

*Proof.* For any instance, fix some $y \in \mathcal{M}$ and let $d_y$ be a metric distance function over $\mathcal{N}$, where for any distinct $i, j \in \mathcal{N}, d_y(i, j) = d^m(i, j) + d^c(i, y) + d^c(j, y)$ (if $i = j$ then we assign $d_y(i, j) = 0$).

We can first show that $d_y$ indeed induces a metric space over $\mathcal{N}$. To show this, we just need to show that the triangle inequality holds, as clearly $d_y$ will meet the other two metric axioms.

For any $i, i', j \in \mathcal{N}$, we should have that $d_y(i, j) \leqslant d_y(i, i') + d_y(i', j)$. By the definition of $d_y$, this is equivalent to saying $d^m(i, j) + d^c(i, y) + d^c(j, y) \leqslant d^m(i, i') + d^c(i, y) + d^c(i', y) + d^m(i', j) + d^c(i', y) + d^c(j, y)$. By the fact that $d^m$ is a metric, we have that $d^m(i, j) \leqslant d^m(i, i') + d^m(i', j)$. Therefore, we just need to show that $d^c(i, y) + d^c(j, y) \leqslant d^c(i, y) + d^c(i', y) + d^c(i', y) + d^c(j, y)$. Clearly, this holds as well, since $d^c(i, y) + d^c(j, y)$ appears on the right-hand side of the inequality.

Algorithm 4 runs the non-centroid greedy capture algorithm on the agents, using the distance metric $d_y$, for every $y \in \mathcal{M}$, then considers that cluster in the original dual metric setting with $y$ as its center, and selects the most cohesive cluster found this way.

For contradiction, assume that Algorithm 4 does not find a 4-approximation of the most cohesive cluster. Let $(S, y)$ be the most cohesive cluster for an instance inputted to Algorithm 4. Define $r = \max_{i \in S} \ell_i(S, y)$ as the maximum loss felt by any agent in $S$. Assume that Algorithm 4 returns the cluster $(C^*, x^*)$, where $r^* = \max_{i \in C^*} \ell_i(C^*, x^*) > 4r$.

Take the distance metric $d_y$, and let $C_y$ be the non-centroid cluster that Algorithm 4 finds while using $d_y$ as its distance function. First, note that $\max_{i \in C_y} \ell_i(C_y, y) \geqslant r^*$. Otherwise, Algorithm 4 would have returned $(C_y, y)$ rather than $(C^*, x)$.

Next, consider the maximum non-centroid loss with respect to $d_y$ for any agent in the member set $S$. We will have $\forall i \in S, \max_{j \in S} d_y(i, j) = \max_{j \in S} \{d^m(i, j) + d^c(i, y) + d^c(j, y)\} \leqslant \max_{j \in S} d^m(i, j) + d^c(i, y) + \max_{j \in S} d^c(j, y) \leqslant r + \max_{j \in S} d^c(j, y)$. For all $j \in S$, we must have that $d^c(j, y) \leqslant r$, which gives us $\max_{j \in S} d_y(i, j) \leqslant 2r$.

Caragiannis et al. [2] prove that in any non-centroid clustering instance, Algorithm 3 achieves a 2-approximation of the most cohesive cluster with respect to non-centroid loss for that instance. Since $C_y$ was found by running Algorithm 3 on $\mathcal{N}$ with distance function $d_y$, $C_y$ must be a 2-approximation to the most cohesive non-centroid cluster in that instance. Particularly, this implies that the maximum non-centroid loss with respect to $d_y$ loss felt by any agent in $C_y$ must be no more than two times the maximum such loss felt by an agent in any member set $C' \subseteq \mathcal{N}$ with $|C'| \geqslant n/k$. With respect to $S$, this implies that $\max_{i,j \in C_y} d_y(i, j) \leqslant 2 \max_{i,j \in S} d_y(i, j) \leqslant 2(2r) = 4r$.

Finally, note that for any member set of agents $C$, the maximum non-centroid loss for that member set felt by any agent $i \in C$ with respect to the metric $d_y$ will be an upper bound on the dual-metric loss agent $i$ will feel under the cluster $(C, y)$. To see this, note that $i$'s non-centroid loss under $C$ with respect to $d_y$ will be equal to $\max_{j \in C} d_y(i, j) = \max_{j \in C}(d^m(i, j) + d^c(i, y) + d^c(j, y)) \geqslant \max_{j \in C} d^m(i, j) + d^c(i, y) = \ell_i(C, y)$. In the case of $C_y$, this tells us that $r^* \leqslant \max_{i \in C_y} \ell_i(C_y, y) \leqslant \max_{i \in C_y} \max_{j \in C_y} d_y(i, j) \leqslant 4r$, which gives a contradiction, concluding the proof. $\square$

Plugging this into Lemma 1 gives us a constant approximation to the core in polynomial time.

**Theorem 3.** *For the dual metric loss, a clustering in the $(3 + 2\sqrt{3})$-core can be computed in polynomial time.*

*Proof.* From Lemma 2, Algorithm 4 computes a 4-MCC cluster. Let Algorithm 1 iteratively use it in Phase 1 and set $\alpha = 4$ in Lemma 1. Then, the growth factor is set to $c = 3 + \sqrt{3}$ and we obtain a core approximation of $\frac{1}{2}(4 + \sqrt{48} + 2) = 3 + 2\sqrt{3}$. Each call to Algorithm 4 runs in polynomial time, and it is clear that the rest of Algorithm 1 also runs in polynomial time. $\square$

## D.2 Weighted Single Metric Loss

**Lemma 3.** *For the weighted single metric loss with parameter $\lambda \in (0, 1]$, Algorithm 5 finds a clustering in the $\frac{2}{\lambda}$-core in polynomial time.*

**Algorithm 5:** Non-Centroid Greedy Capture With Greedy Centroid Selection
**Input:** Set of agents $\mathcal{N}$, set of feasible centers $\mathcal{M}$, metric $d$, number of clusters $k$
**Output:** Clustering $\mathcal{X} = \{(C_1, x_1), \ldots, (C_k, x_k)\}$
1 Initialize: $\mathcal{N}' \leftarrow \mathcal{N}, \mathcal{X} \leftarrow \emptyset, \delta \leftarrow 0$;
    // Uncaptured agents remain
2 **while** $\mathcal{N}' \neq \emptyset$ **do**
3     Increase $\delta$ smoothly;
        // Non-centroid ball found
4     **if** $\exists i \in \mathcal{N}', C \subseteq \mathcal{N}'$ s.t. $|C| \geqslant \min(|\mathcal{N}'|, n/k)$ *and* $d(i, i') \leqslant \delta$ *for all* $i' \in C$ **then**
5         $x \leftarrow \arg\min_{x \in \mathcal{M}} d(i, x)$;    // Centroid closest to the ball's center
6         $\mathcal{X} \leftarrow \mathcal{X} \cup \{(C, x)\}$;
7         $\mathcal{N}' \leftarrow \mathcal{N}' \setminus C$;

8 **return** $\mathcal{X}$

---

*Proof.* For contradiction, assume this is false, and there exists some instance such that the greedy clustering solution $\mathcal{X} = \{(C_1, x_1), \ldots, (C_k, x_k)\}$ is not in the $\frac{2}{\lambda}$-core. This means there exists some deviating cluster $(S, y)$ such that for all $i \in S$, $\frac{2}{\lambda} \cdot \ell_i(S, y) < \ell_i(\mathcal{X})$.

Let $i = \arg\min_{i' \in S} \mathcal{X}(i')$ be the first agent from $S$ captured by Algorithm 5. Let $\mathcal{X}(i) = t$. Let $j \in C_t$ be the agent that "captured" the cluster $C_t$ (meaning that at the time when $C_t$ was captured by the algorithm, $d(i', j) \leqslant \delta$ for all $i' \in C_t$ was true), and let $r_t = \max_{i \in C_t} d(i, j)$ be the value of $\delta$ at the point when $C_t$ was captured.

Due to the triangle inequality, we have that for any $i' \in C_t, d(i, i') \leqslant d(i, j) + d(j, i') \leqslant 2r_t$. With this, we have that $\ell_i(\mathcal{X}) \leqslant (1 - \lambda)d(i, x_t) + \lambda 2r_t$.

Note that there must be some $i' \in S$ with $d(i, i') \geqslant r_t$. If this were not the case, then Algorithm 5 would have captured $S$ prior to capturing $C_t$, with $i$ being the "capturing" agent of $S$. Thus, we also have that $\ell_i(S, y) \geqslant (1 - \lambda)d(i, y) + \lambda r_t$.

Due to the way that Algorithm 5 greedily selects a center for each cluster, it must be the case that $d(j, y) \geqslant d(j, x_t)$, since otherwise, $y$ would have been selected as the center for $C_t$ rather than $x_t$. From the triangle inequality, we have that $d(i, x_t) \leqslant d(i, j) + d(j, x_t) \leqslant r + d(j, y) \leqslant r + d(i, j) + d(i, y) \leqslant 2r + d(i, y)$. This allows us to conclude that $d(i, x_t) \leqslant 2r + d(i, y)$

The above inequality allows us to bound $\ell_i(\mathcal{X}) \leqslant (1 - \lambda)d(i, x_t) + \lambda 2r \leqslant (1 - \lambda)(2r + d(i, y)) + \lambda 2r = 2r + (1 - \lambda)d(i, y)$. This means that $i$ improves their loss by deviating by at most a factor of:

$$\frac{2r_t + (1 - \lambda)d(i, y)}{(1 - \lambda)d(i, y) + \lambda r_t} = \frac{(2 - \lambda)r_t}{(1 - \lambda)d(i, y) + \lambda r_t} + 1 \leqslant \frac{(2 - \lambda)r_t}{\lambda r_t} + 1 = \frac{2}{\lambda}$$

This gives the desired contradiction. $\qquad\square$

**Lemma 4.** *For the weighted single metric loss with parameter $\lambda \in [0, 1)$, Algorithm 6 finds a clustering in the $\frac{\sqrt{2\lambda - 11\lambda^2 + 13} + 3 - \lambda}{2 - 2\lambda}$-core in polynomial time.*

*Proof.* When running Algorithm 6, we set the growth factor $c = \frac{q}{\lambda}$, where $q = \frac{\sqrt{2\lambda - 11\lambda^2 + 13} + 5\lambda - 1}{6}$.

Let $\mathcal{X} = \{(C_1, x_1), \ldots, (C_k, x_k)\}$ be the clustering returned by Algorithm 6. Further, let $\widehat{\mathcal{X}} = \{(\widehat{C}_1, x_1), \ldots, (\widehat{C}_k, x_k)\}$ be the clustering found by Algorithm 6 after it has completed phase 1.

First, we will show that for any agent $i \in N$, if $\widehat{\mathcal{X}}(i) = t$, then $\ell_i(\mathcal{X})$ will be no more than $(1 - \lambda)d(i, x_t) + \lambda(d(i, x_t) + \frac{q}{\lambda}r_t) = d(i, x_t) + q \cdot r_t$. This can be seen by analyzing how the second phase of the algorithm works.

In phase 2 of Algorithm 6, each agent $i \in \mathcal{N}$ can choose to switch from the cluster they were originally assigned to by the algorithm, to a new cluster. However, they can only switch to a cluster $(C_{t'}, x_{t'})$ if $d(i, x_{t'}) \leqslant cr_{t'}$. In the algorithm, $c$ is called the "growth factor", as the second phase can

---

**Algorithm 6:** Semi-Ball-Growing Algorithm

---

**Input:** Set of agents $\mathcal{N}$, set of feasible centers $\mathcal{M}$, distance metric $d$, parameter $\lambda \in [0, 1]$,
      number of clusters $k$, growth factor $c \geqslant 1$
**Output:** Clustering $\mathcal{X} = \{(C_1, x_1), \ldots, (C_k, x_k)\}$

**1** Initialize: $\mathcal{N}' \leftarrow \mathcal{N}, \widehat{\mathcal{X}} \leftarrow \emptyset, \delta \leftarrow 0, t \leftarrow 0$;
   // Phase 1:  Build a tentative clustering via centroid greedy capture
**2 while** $\mathcal{N}' \neq \emptyset$ **do**                                 // Uncaptured agents remain
**3**    |  Increase $\delta$ smoothly;
**4**    |  **if** $\exists x_t \in \mathcal{M}, \widehat{C}_t \subseteq \mathcal{N}'$ s.t. $|\widehat{C}_t| \geqslant \min(|\mathcal{N}'|, {}^n\!/_k)$ and $d(i, x_t) \leqslant \delta$ for all $i \in \widehat{C}_t$ **then**
**5**    |    |  $\widehat{\mathcal{X}} \leftarrow \widehat{\mathcal{X}} \cup \{(\widehat{C}_t, x_t)\}$;
**6**    |    |  $r_t \leftarrow \delta$;
**7**    |    |  $\mathcal{N}' \leftarrow \mathcal{N}' \setminus \widehat{C}_t$;
**8**    |    |  $t \leftarrow t + 1$

   // Phase 2:  Selectively allow agents to switch clusters
**9** $\forall t \in [k], C_t \leftarrow \widehat{C}_t$;
**10 for** $i \in \mathcal{N}$ **do**
   |  // Allowed cluster with the best upper bound on loss
**11**    |  $t' \leftarrow \arg\min_{t \in [k]: d(i, x_t) \leqslant c \cdot r_t} \{(1 - \lambda) \cdot d(i, x_t) + c \cdot \lambda \cdot 2r_t\}$;
**12**    |  $t^* \leftarrow \widehat{\mathcal{X}}(i)$;
   |  // Switch if it improves the upper bound on loss
**13**    |  **if** $(1 - \lambda) \cdot d(i, x_{t'}) + c \cdot \lambda \cdot 2r_{t'} < d(i, x_{t^*}) + c \cdot \lambda \cdot r_{t^*}$ **then**
**14**    |    |  $C_{t^*} \leftarrow C_{t^*} \setminus \{i\}$;
**15**    |    |  $C_{t'} \leftarrow C_{t'} \cup \{i\}$;

**16 return** $\mathcal{X} = \{(C_1, x_1), \ldots, (C_k, x_k)\}$

---

be thought of as the radius of each of the cluster balls increasing by a factor of $c$, then each agent choosing to change to a new cluster that they are within the radius of. Also note that each agent judges whether some new cluster $(C_{t'}, x_{t'})$ will provide it less loss than its current cluster based on the *worst-case* loss $i$ could possibly feel after switching to $(C_{t'}, x_{t'})$. This is reflected in the term $c \cdot \lambda \cdot 2r_t$ on Line 11 of Algorithm 6. Due to the fact that the non-centroid part of each agent's loss function will change based on which clusters other agents choose to switch into during the second phase of the algorithm, each agent will only switch to a new cluster if, regardless of how other agents switch, their loss under that new cluster is still guaranteed to be less than their current loss.

For any $i \in \mathcal{N}$ with $\widehat{\mathcal{X}}(i) = t$, if $i$ were to not switch to a new cluster in the second phase of the algorithm, then we can show that its final loss will be less than $d(i, x_t) + q \cdot r_t$. To see this, assume $i$ does not switch clusters in phase 2, and thus $\mathcal{X}(i) = t$. Because we know that $i$ did not switch, it must be the case that the distance from $i$ to the centroid of its final cluster is $d(i, x_t) \leqslant r_t$. Further, from the way that agents were allowed to switch clusters in the second phase, we know that all for any $j \in C_t$, we must have $d(j, x_t) \leqslant c \cdot r_t$. By the triangle inequality, this means that for $j = \max_{j \in C_t} d(j, i)$, the agent from $C_t$ that is farthest away from $i$, we must have $d(i, j) \leqslant d(i, x_t) + d(j, x_t) \leqslant d(i, x_t) + c \cdot r_t$.

Putting these together and plugging in that $c = \frac{q}{\lambda}$ we have that:

$$\ell_i(\mathcal{X}) = (1 - \lambda) \cdot d(i, x_t) + \lambda \cdot d(i, j) \leqslant (1 - \lambda) \cdot d(i, x_t) + \lambda \cdot (d(i, x_t) + c \cdot r_t) = d(i, x_t) + q \cdot r_t$$

Similarly, we can bound $i$'s loss if they do choose to switch clusters in phase 2. Assume now that $i$ does switch such that after the second phase of the algorithm $\mathcal{X}(i) = t'$.

By similar logic as above, we can upper bound $i$'s loss as:

$$\ell_i(\mathcal{X}) = (1 - \lambda)d(i, x_{t'}) + \lambda \ell_i^m(\mathcal{X}) \leqslant (1 - \lambda)d(i, x_{t'}) + \lambda 2cr_{t'} = (1 - \lambda)d(i, x_{t'}) + 2qr_{t'}$$

Here, the second transition follows from the fact that every agent in $C_{t'}$ is at most a distance of $cr_{t'}$ away from $x_{t'}$. This along with a similar triangle inequality argument as above allows us to bound $\ell_i^m(C_{t'}, x_{t'}) \leqslant 2cr_{t'}$. Notice that this is worse than the bound we were able to establish on $i$'s non-centroid loss in the case when it did not switch. This is because when $i$ did not switch, we knew that $d(i, x_t) \leqslant r_t$, where in the case where $i$ did switch, all we know is the weaker bound of $d(i, x_{t'}) \leqslant c \cdot r_t$.

From line 13 of Algorithm 6, we can see that an agent $i$ will only switch from their first phase cluster to a new cluster if the upper-bounded loss from this new cluster is less than the upper-bounded loss of their original cluster. Thus, no matter what $i$ does during the second phase, we can guarantee that their loss will not be more than $d(i, x_t) + q \cdot r_t$.

With this bound on each agent's loss established, we can prove the required core guarantee. Assume for contradiction that this is false, and that Algorithm 6 produces a clustering $\mathcal{X}$ such that there is some deviating cluster $(S, y)$ with $S \subseteq \mathcal{N}, |S| \geqslant \lceil \frac{n}{k} \rceil$, and $y \in \mathcal{M}$ with $\frac{\sqrt{2\lambda - 11\lambda^2 + 13} + 3 - \lambda}{2 - 2\lambda} \ell_i(S, y) < \ell_i(\mathcal{X})$ for all $i \in S$.

Let $i = \min_{i \in S} \widehat{\mathcal{X}}(i)$ be the agent from $S$ that was captured first by phase 1 of the algorithm, and let $\widehat{\mathcal{X}}(i) = t$. We know that $d(i, x_t) \leqslant r_t$. Let $j = \arg\max_{j \in S} d(j, y)$ be the agent in $S$ that is farthest away from $y$. We know that $d(j, y) \geqslant r_t$ must be true, otherwise, the first phase of the algorithm would have captured $(S, y)$ prior to capturing $(\widehat{C}_t, x_t)$.

First, we will consider the case where $d(j, x_t) \leqslant c \cdot r_t$. This means that in the phase 2 of the algorithm, $j$ will have the opportunity to switch to the $t$-th cluster. The maximum loss that $j$ could feel under the cluster centered at $x_t$ is $(1 - \lambda)d(j, x_t) + 2qr_t$, and since $j$ will switch to the cluster with the lowest upper-bounded loss, this means that $j$ deviating to $(S, y)$ can improve its loss by at most a factor of $\frac{(1-\lambda)d(j,x_t)+2qr_t}{(1-\lambda)d(j,y)}$. Note that this can be upper bounded as

$$
\begin{aligned}
\frac{(1 - \lambda)d(j, x_t) + 2qr_t}{(1 - \lambda)d(j, y)} &\leqslant \frac{(1 - \lambda)(d(j, y) + d(i, y) + d(i, x_t)) + 2qr_t}{(1 - \lambda)d(j, y)} \\
&\leqslant \frac{(1 - \lambda)d(j, y) + (1 - \lambda)(d(i, y) + r_t) + 2qr_t}{(1 - \lambda)d(j, y)} \\
&\leqslant \frac{(1 - \lambda)r_t + (1 - \lambda)(d(i, y) + r_t) + 2qr_t}{(1 - \lambda)r_t} \\
&= \frac{(2(1 - \lambda) + 2q)r_t + (1 - \lambda)d(i, y)}{(1 - \lambda)r_t} \\
&\leqslant \frac{2(1 - \lambda) + 2q}{(1 - \lambda)} + \frac{d(i, y)}{r_t}. \quad (2)
\end{aligned}
$$

For $i$, we must have that $\ell_i^m(S, y) \geqslant (1 - \lambda)d(i, y)$, and $\ell_i^m(\mathcal{X})$ cannot be more than $i$'s upper-bounded loss under $(\widehat{C}_t, x_t)$, meaning that $i$ deviating cannot improve its loss by more than a factor of

$$
\frac{d(i, x_t) + q \cdot r_t}{(1 - \lambda)d(i, y)} \leqslant \frac{(1 + q)r_t}{(1 - \lambda)d(i, y)}. \quad (3)
$$

Combining Equations (2) and (3), we get the following bound:

$$
\min\left\{ \frac{1 + q}{1 - \lambda} \cdot \frac{r_t}{d(i, y)}, \frac{2(1 - \lambda) + 2q}{(1 - \lambda)} + \frac{d(i, y)}{r_t} \right\} \leqslant \max_{z \geqslant 0} \min\left\{ \frac{1 + q}{1 - \lambda}z, \frac{2(1 - \lambda) + 2q}{(1 - \lambda)} + \frac{1}{z} \right\}
$$

$$
= \frac{\sqrt{\lambda^2 - 3\lambda(q + 1) + q^2 + 3q + 2} - \lambda + q + 1}{1 - \lambda},
$$

where the last transition can be derived by equating the two expressions in $z$ and solving the resulting quadratic equation in $z$. Substituting the chosen value of $q$ and simplifying gives us the following bound:

$$
\frac{\sqrt{2\lambda - 11\lambda^2 + 13} + 3 - \lambda}{2 - 2\lambda}.
$$

This yields a contradiction. This shows that it must be true that $d(j, x_t) > c \cdot r_t$, meaning that $j$ cannot switch to the $t$-th cluster during the second phase of Algorithm 6.

Using this along with the triangle inequality, we can say that $\frac{q}{\lambda} r_t < d(j, x_t) \leqslant d(j, i) + d(i, x_t) \leqslant d(j, i) + r_t$. Rearranging we get that $d(i, j) > (\frac{q}{\lambda} - 1)r_t = (\frac{q - \lambda}{\lambda})r_t$.

With this, we can say that $\ell_i(S, y) \geqslant \lambda d(i, j) > \lambda \frac{q - \lambda}{\lambda} r_t = (q - \lambda)r_t$.

Combining this with the fact that $\ell_i(\mathcal{X}) \leqslant d(i, x_t) + q \cdot r_t \leqslant (q + 1)r_t$, we get that in this case, $i$ deviating to $(S, y)$ cannot improve its loss by more than a factor of $\frac{q+1}{q-\lambda}$.

Substituting in our chosen value of $q$ and simplifying once again gives us:

$$\frac{\sqrt{2\lambda - 11\lambda^2 + 13} + 3 - \lambda}{2 - 2\lambda}$$

which is a contradiction, and concludes the proof. $\qquad \square$

**Lemma 9.** *For the weighted single metric loss with parameter $\lambda \in [0, 1]$, no clustering algorithm can achieve a core approximation better than $\frac{\sqrt{\lambda^2 - 2\lambda + 5} - \lambda + 1}{2}$.*

*Proof.* We will analyze the instance in Figure 1, with $p = \frac{\sqrt{\lambda^2 - 2\lambda + 5} + \lambda - 1}{2\lambda}$. We will limit our analysis to the section of the instance containing $\{a, b, c\}$, where, since $k = 3$, we can assume without loss of generality that one cluster center is placed.

If the cluster $(\{a, b, c\}, b)$ is initially selected, then $a$ will have a loss of $(1 - \lambda)d(a, b) + \lambda d(a, c) = (1 - \lambda) + \lambda p$, and $b$ will have a loss of $(1 - \lambda)0 + \lambda p = \lambda p$. If $a$ and $b$ deviate to the cluster $(\{a, b\}, b)$, then they will improve their loss by a factor of $(1 - \lambda) + \lambda p$ and $p$ respectively. This deviation will lead to a violation of core with a factor of $(1 - \lambda) + \lambda p = \frac{\sqrt{\lambda^2 + 2\lambda + 5} - \lambda + 1}{2}$. The same argument can show that $(\{a, b, c\}, a)$ would also not work as a cluster, and clearly, $(\{a, b, c\}, c)$ would also not work, as $a$ and $b$'s losses would only get worse then before.

If the cluster $(\{a, b\}, b)$ is initially selected, then note that $a$ will have a loss of $(1 - \lambda)d(a, b) + \lambda d(a, b) = 1$. In contrast, under the clustering $(\{a, c\}, a)$, $a$ will have a loss of $(1 - \lambda)0 + \lambda d(a, c) = \lambda p$. This means that by deviating, $a$ will decrease its loss by a factor of $\frac{1}{\lambda p} = \frac{2}{\sqrt{\lambda^2 - 2\lambda + 5} + \lambda - 1} = \frac{\sqrt{\lambda^2 + 2\lambda + 5} - \lambda + 1}{2}$. Since we assume that only one point from $\{a, b, c\}$ is selected as a center, this deviation must also improve $c$'s loss by an infinite factor. Note that the same analysis holds if $a$ was selected as the centroid in the initial cluster rather than $b$. Again, clearly the center $c$ would not work with this cluster either, as it would just make both agents no better off.

Next, note that the cluster $(\{a, c\}, a)$ cannot be selected. $a$ would have a loss of $(1 - \lambda)0 + \lambda p = \lambda p$. But under the clustering $(\{a, b\}, a)$ $a$'s loss would be $(1 - \lambda)0 + \lambda = \lambda$, so deviating would improve $a$'s loss by a factor of $p \geqslant \frac{\sqrt{\lambda^2 + 2\lambda + 5} - \lambda + 1}{2}$. $b$ would also improve their loss by an infinite factor. Clearly trying this cluster with any other center would not help, as it would only make $a$'s initial loss worse. This argument can also be used to show that the cluster with member set $\{b, c\}$ will not work.

This eliminates the possibility of any cluster of the 3 agents of size greater than 1. Finally, we cannot have more than 2 of these agents involved in a singleton cluster, as since $k = 3$, all the agents on the other side would have to all belong to the same cluster, and we could apply a similar argument as the second paragraph of this proof to find a deviating set there. $\qquad \square$

**Lemma 10.** *For the weighted single metric loss with parameter $\lambda \in [0, 1]$, no clustering algorithm can achieve a core approximation better than $\frac{2(1 - \lambda)}{2\lambda + 1}$.*

*Proof.* Take the instance from Claim 1 of Chen et al. [1]. $\mathcal{N} = \{a_1, \ldots, a_6\}, \mathcal{M} = \{y_1, \ldots, y_6\}, k = 3$. The distances between agents are given as $d(a_1, a_2) = d(a_1, a_3) = d(a_2, a_3) = 3$ and $d(a_4, a_5) = d(a_4, a_6) = d(a_5, a_6) = 3$, with all other pairs having distance $\infty$. The distances between agents and centers are given by the below table (which can also be seen in claim 1 of Chen et al. [1]).

| | $y_1$ | $y_2$ | $y_3$ | $y_4$ | $y_5$ | $y_6$ |
|---|---|---|---|---|---|---|
| $a_1$ | 4 | 1 | 2 | $\infty$ | $\infty$ | $\infty$ |
| $a_2$ | 2 | 4 | 1 | $\infty$ | $\infty$ | $\infty$ |
| $a_3$ | 1 | 2 | 4 | $\infty$ | $\infty$ | $\infty$ |
| $a_4$ | $\infty$ | $\infty$ | $\infty$ | 4 | 1 | 2 |
| $a_5$ | $\infty$ | $\infty$ | $\infty$ | 2 | 4 | 1 |
| $a_6$ | $\infty$ | $\infty$ | $\infty$ | 1 | 2 | 4 |

Chen et al. [1] show that in this instance, it is impossible to achieve a centroid core approximation better than 2.

Fix some clustering $\mathcal{X} = \{(C_1, x_1), (C_2, x_2), (C_3, x_3)\}$. From Chen et al. [1], there must be some pair of agents that would want to deviate in the centroid world for a core improvement of at least 2. More specifically, since there are 3 clusters, and the instance consists of two "groups" of agents/centers, which are separated by a distance of $\infty$, one of the groups must have at most 1 center from that group selected as one of the centers in $\mathcal{X}$.

Without loss of generality, assume that the group $\{a_1, a_2, a_3\}$ only has one center from their group selected by $\mathcal{X}$, and assume $y_1$ is that center (because of the symmetry of the instance, we could pick either group, and any center of that group, and an identical argument would hold). In this scenario, the other two centers from $\mathcal{X}$ are distance $\infty$ away from $a_1$ and $a_2$. $a_1$ and $a_2$ are either in the cluster with $y_1$, or in a cluster with a center farther away from them than $y_1$. Therefore, we must have that $\ell_{a_1}(\mathcal{X}) \geqslant (1-\lambda)d(a_1, y_1) = 4(1-\lambda)$ and $\ell_{a_2}(\mathcal{X}) \geqslant (1-\lambda)d(a_2, y_1) = 2(1-\lambda)$.

Now consider the deviating cluster $(S = \{a_1, a_2\}, y_3)$. $a_1$ and $a_2$ will have the following losses under that cluster:

$$\ell_{a_1}(S, y_3) = \lambda d(a_1, a_2) + (1-\lambda)d(a_1, y_3) = 3\lambda + 2(1-\lambda)$$
$$\ell_{a_2}(S, y_3) = \lambda d(a_1, a_2) + (1-\lambda)d(a_2, y_3) = 3\lambda + 1(1-\lambda)$$

These, along with our lower bounds on the agents' loss under $\mathcal{X}$ allow us to conclude that $a_1$ would be able to make an improvement of at least $\frac{4(1-\lambda)}{3\lambda+2(1-\lambda)}$ by deviating, and $a_2$ would see an improvement of at least $\frac{2(1-\lambda)}{3\lambda+1(1-\lambda)}$.

It can easily be verified that $\frac{4(1-\lambda)}{3\lambda+2(1-\lambda)} \geqslant \frac{2(1-\lambda)}{3\lambda+1(1-\lambda)}$ for all $\lambda \in [0, 1]$, meaning that this would lead to a core violation of $\frac{2(1-\lambda)}{3\lambda+1(1-\lambda)} = \frac{2(1-\lambda)}{2\lambda+1}$, completing the proof. $\qquad \square$

With the above two lower-bounds, we can make the following general statement.

**Theorem 5.** *For the weighted single metric loss with parameter $\lambda \in [0, 1]$, no clustering algorithm can achieve a core approximation better than $\max\{\frac{\sqrt{\lambda^2-2\lambda+5}-\lambda+1}{2}, \frac{2(1-\lambda)}{2\lambda+1}\}$.*

*Proof.* The result for $\lambda \in (0, 1)$ follows directly from Lemmas 9 and 10. The edge cases follow from known lower bounds from the centroid (lower bound of 2 [1]) and non-centroid (lower bound of 1 [2]) worlds. $\qquad \square$

## E   Missing Proofs from Section 5

First, we provide the missing proof of Theorem 6, which shows that Algorithm 7 yields an $\alpha$-FJR clustering, guaranteeing the existence of an (exact) FJR clustering for arbitrary loss functions and the polynomial-time computability of a 4-FJR clustering for the dual metric loss.

**Theorem 6.** *For arbitrary losses and $\alpha \geqslant 1$, Algorithm 7, which iteratively finds an $\alpha$-approximate most cohesive cluster, returns an $\alpha$-FJR clustering. Hence, a 1-FJR clustering exists for arbitrary losses and, due to Lemma 2, a 4-FJR clustering can be computed in polynomial time for the dual metric loss.*

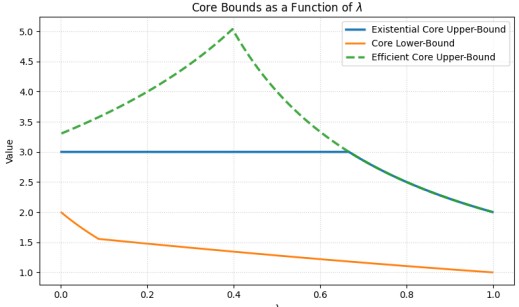

Figure 2: Single Metric Core Bounds Visualized as $\lambda$ varies from 0 to 1. For exact bounds, see Table 1

---

**Algorithm 7:** Iterative $\alpha$-MCC Clustering

---

**Input:** Set of agents $\mathcal{N}$, set of feasible centers $\mathcal{M}$, agents' loss functions $(\ell_i)_{i \in \mathcal{N}}$, number of centers $k$, approximation factor $\alpha \geqslant 1$;

**Output:** Clustering $\mathcal{X} = \{(C_1, x_1), \ldots, (C_k, x_k)\}$

1 Initialize: $\mathcal{N}' \leftarrow \mathcal{N}$, $\mathcal{X} \leftarrow \emptyset$;
   // Uncaptured agents remain
2 **while** $\mathcal{N}' \neq \emptyset$ **do**
3    $(C, x) \leftarrow$ An $\alpha$-MCC cluster (with threshold $n/k$) from $\mathcal{N}', \mathcal{M}$ ;   // Find an $\alpha$-MCC cluster
4    $\mathcal{X} \leftarrow \mathcal{X} \cup \{(C, x)\}$;   // Add it to the clustering
5    $\mathcal{N}' \leftarrow \mathcal{N}' \setminus C$;   // Mark its agents as captured
6 **return** $\mathcal{X}$

---

*Proof.* For contradiction, assume that Algorithm 7 does not always yield an $\alpha$-FJR clustering. Then, there exists an instance in which the clustering $\mathcal{X} = \{(C_1, x_1), \ldots, (C_k, x_k)\}$ returned by Algorithm 7 admits a deviating cluster $(S, y)$ such that for all $i \in S$,

$$\alpha \cdot \ell_i(S, y) < \min_{j \in S} \ell_j(\mathcal{X}). \tag{4}$$

Let $i^* \in \arg\min_{i \in S} \mathcal{X}(i)$ be the first agent in $S$ that was "captured" (i.e., removed from $\mathcal{N}'$) by the algorithm, ties broken arbitrarily. Let $t = \mathcal{X}(i^*)$. Then, from Equation (4), we get that for all $i \in S$,

$$\alpha \cdot \ell_i(S, y) < \min_{j \in S} \ell_j(\mathcal{X}) \leqslant \ell_{i^*}(\mathcal{X}) \leqslant \max_{i \in C_t} \ell_i(\mathcal{X}) = \max_{i \in C_t} \ell_i(C_t, x_t).$$

Hence, $\alpha \cdot \max_{i \in S} \ell_i(S, y) < \max_{i \in C_t} \ell_i(C_t, x_t)$. Since all agents in $S$ were uncaptured prior to $(C_t, x_t)$ being added to $\mathcal{X}$, this contradicts $(C_t, x_t)$ being an $\alpha$-MCC cluster at that time. This concludes the proof that Algorithm 7 always yields an $\alpha$-FJR clustering.

Since an MCC cluster exists by definition, plugging in $\alpha = 1$ yields the existence of an FJR clustering. And for the dual metric loss, since a 4-MCC cluster can be computed efficiently via Algorithm 4 (Lemma 2), we obtain that a 4-FJR clustering can be computed in polynomial time. $\square$

Next, we provide the missing proof of Theorem 7, which shows that such a simultaneous approximation is impossible when using different centroid and non-centroid metrics.

**Theorem 7.** *For any $\alpha \geqslant 1$ and $\beta \geqslant 1$, there exists an instance in which no clustering is both $\alpha$-FJR with respect to the centroid loss and $\beta$-FJR with respect to the non-centroid loss, when the two loss functions are allowed to use different metrics.*

*Proof.* Consider an instance with a set of 6 agents $\mathcal{N} = \{a, b, c, d, e, f\}$, a set of two feasible cluster centers $\mathcal{M} = \{x_1, x_2\}$, and $k = 3$.

*Non-centroid metric:* Let $d^m(a, b) = d^m(c, d) = d^m(e, f) = 0$, and let the non-centroid distance between every other pair of agents be $\infty$.[3]

---

[3]Here, we use $\infty$ for the ease of exposition, but it can be replaced by a suitably large number.

*Centroid metric:* Let $d^c(a, x_1) = d^c(c, x_1) = d^c(e, x_1) = 0$, let $d^c(b, x_2) = d^c(d, x_2) = d^c(f, x_2) = 0$, and let the centroid distance between every other agent-center pair be $\infty$.

*Finite non-centroid FJR approximation:* First, note that the only clustering that leads to a finite FJR approximation with respect to the non-centroid loss is the one that partitions the set of agents as $\mathcal{X} = \{\{a, b\}, \{c, d\}, \{e, f\}\}$.

To see this, consider any $C \in \mathcal{X}$. If $C$ is "broken" (i.e., not a cluster by itself), then at least one member of $C$ must be a singleton cluster; the only other possibility would be that $C$ is a strict subset of a cluster, in which case $C$ can deviate, improving both its members from infinite loss to zero loss, yielding infinite FJR approximation.

Hence, every unbroken pair in $\mathcal{X}$ forms a cluster of size two, and (at least) one member from every broken pair in $\mathcal{X}$ forms a singleton cluster. Since there are three pairs in $\mathcal{X}$, this already uses up $k = 3$ clusters, leaving the other members from the broken pairs in $\mathcal{X}$ not included in the partition, which is impossible. Hence, there must not be any broken pairs, which shows that the partition must be $\mathcal{X} = \{\{a, b\}, \{c, d\}, \{e, f\}\}$.

*Finite centroid FJR approximation:* Now, let us try to assign a center to each cluster in $\mathcal{X}$ in a way that yields a finite approximation with respect to the centroid loss. Since there are only two feasible centers $x_1$ and $x_2$, at least one of them must be assigned to at most one of the clusters in $\mathcal{X}$. Without loss of generality, assume this center is $x_1$. There are three agents, $a$, $c$, and $e$, which are at distance $0$ away from $x_1$ under the centroid metric. However, they are in different clusters. Hence, at least two of them must be assigned to $x_2$, and can deviate together with $x_1$ and improve both their centroid losses from non-zero to zero, resulting in an infinite FJR approximation with respect to the centroid loss. If we had assumed $x_2$, we could have made the same argument with agents $b$, $d$, and $f$.

This shows that there is no clustering that achieves finite FJR approximations with respect to both the centroid and non-centroid losses, concluding the proof. $\square$

Next, we provide the missing proof of Theorem 8, which shows that simultaneous approximation is possible if both centroid and non-centroid losses use the same underlying metric. Since our algorithm (Algorithm 5) builds on non-centroid greedy capture, its FJR approximation with respect to the non-centroid loss is borrowed easily; the key difficulty is establishing its FJR approximation with respect to the centroid loss.

**Lemma 11.** *Algorithm 5, given metric d, yields a clustering that is 5-FJR with respect to the centroid loss induced by d.*

*Proof.* For contradiction, assume this is false. Then, there exists an instance in which Algorithm 5 returns a clustering $\mathcal{X} = \{(C_1, x_1), \ldots, (C_k, x_k)\}$ that admits a deviating cluster $(S, y)$ such that

$$\forall q \in S : 5 \cdot d(q, y) < \min_{p \in S} \ell_p^c(\mathcal{X}). \tag{5}$$

Let $i \in S$ be the agent from $S$ that was captured first by Algorithm 5. Denote the cluster containing agent $i$ as $(C, x) \triangleq (C_{\mathcal{X}(i)}, x_{\mathcal{X}(i)})$ for brevity. Because Algorithm 5 runs non-centroid greedy capture to create the partition of the set of agents, $C$ must have been captured by a ball centered at some agent $j \in C$. Note that the center $x$ selected by Algorithm 5 for $C$ must be the closest feasible center to agent $j$. Let $r = \max_{i' \in C} d(j, i')$ be the radius of the ball centered at $j$ when it captured $C$.

First, we show that $d(i, x) \leqslant 2r + d(i, y)$. To see this, note that

$$d(i, x) \leqslant d(i, j) + d(j, x) \leqslant d(i, j) + d(j, y) \leqslant d(i, j) + d(i, j) + d(i, y) \leqslant 2r + d(i, y),$$

where the first and third inequalities use the triangle inequality, the second inequality uses the fact that $x$ is the closest center to agent $j$ (hence, $d(j, x) \leqslant d(j, y)$), and the last inequality follows from the definition of $r$.

Now, from Equation (5), we have that

$$\forall q \in S : 5 \cdot d(q, y) < \min_{p \in S} \ell_p^c(\mathcal{X}) \leqslant \ell_i^c(\mathcal{X}) = d(i, x) \leqslant 2r + d(i, y).$$

In particular, substituting $q = i$, we get that $5 \cdot d(i, y) < 2r + d(i, y)$, which yields $d(i, y) < r/2$. Then, for every other agent $q \in S$, we have that $5 \cdot d(q, y) < 2r + d(i, y) < 5r/2$, which yields $d(q, y) < r/2$.

By the triangle inequality, we have that, for all $p, q \in S$, $d(p,q) \leqslant d(p,y) + d(q,y) < r/2 + r/2 = r$. But this is a contradiction because this would imply that in the execution of Algorithm 5, the ball centered at any agent in $S$ would have captured $S$ before the ball centered at agent $j$ captured $C$. $\quad\square$

The proof of our simultaneous FJR approximation guarantee (Theorem 8) is now straightforward.

**Theorem 8.** *Algorithm 5, given metric $d$, finds, in polynomial time, a clustering that is $5$-FJR with respect to the centroid loss and $2$-FJR with respect to the non-centroid loss, both defined using $d$.*

*Proof.* $5$-FJR with respect to the centroid loss $\ell^c$ is established in Lemma 11.

Caragiannis et al. [2] show that non-centroid greedy capture achieves $2$-FJR with respect to the non-centroid loss $\ell^m$. It is easy to see that because Algorithm 5 runs non-centroid greedy capture and uses the partition $(C_1, \ldots, C_k)$ of the set of agents $\mathcal{N}$ that it produces, it retains this $2$-FJR guarantee with respect to the non-centroid loss $\ell^m$. $\quad\square$

# F   Experiments

In this section, our goal is to empirically compare proportionally fair algorithms designed in this work to classical clustering methods, in terms of both proportional fairness metrics and traditional metrics that the classical methods optimize.

Because the classical methods take only a single distance metric $d$ as input, we limit our empirical analysis to weighted single metric loss, which depends on a single metric $d$ along with a parameter $\lambda \in [0,1]$.

**Algorithms.** We evaluate four algorithms.

- *GC*: Algorithm 5, which runs non-centroid greedy capture followed by greedy centroid selection, and attains $\frac{2}{\lambda}$-core (Lemma 3). We denote it by 'GC'.

- *SemiBall*: Algorithm 6, which mimics centroid greedy capture but with limited cluster switching, and attains $\frac{\sqrt{2\lambda - 11\lambda^2 + 13} + 3 - \lambda}{2 - 2\lambda}$-core (Lemma 4). We denote it by 'SemiBall'.

- *$k$-means++*: This classical algorithm approximately optimizes the $k$-means objective, $\sum_{i \in \mathcal{N}} d(i, \mathcal{X}(i))^2$.

- *$k$-medoids:* This classical algorithm approximately optimizes the $k$-medoids objective, $\sum_{i \in \mathcal{N}} d(i, \mathcal{X}(i))$.

The $k$-means++ and $k$-medoids implementations are based on the `Scikit-learn` package in Python.[4]

It is worth remarking that three of the four algorithms—GC, $k$-means++, and $k$-medoids—take only the distance metric $d$ as input and not the parameter $\lambda$. Hence, the clustering returned by these methods is independent of $\lambda$, although its approximations to proportional fairness metrics would depend on $\lambda$.

**Datasets.** We use three datasets from the UCI Machine Learning Repository [29]: *Iris*, *Pima Indians Diabetes*, and *Adult*. These are the three datasets used by Caragiannis et al. [2] for their experiments with non-centroid clustering.

- The *Iris* dataset contains 150 data points with measurements of sepal and petal dimensions across three species of iris flowers.

- The (Pima Indians) *Diabetes* dataset includes health-related indicators such as insulin dosage, glucose level, and number of pregnancies.

- The *Adult* dataset consists of 48,842 instances, with both categorical features (such as race and education) and numerical features (such as age and hours worked per week). It is typically used for the classification task of predicting whether an individual's income

---

[4]Scikit-learn: https://scikit-learn.org

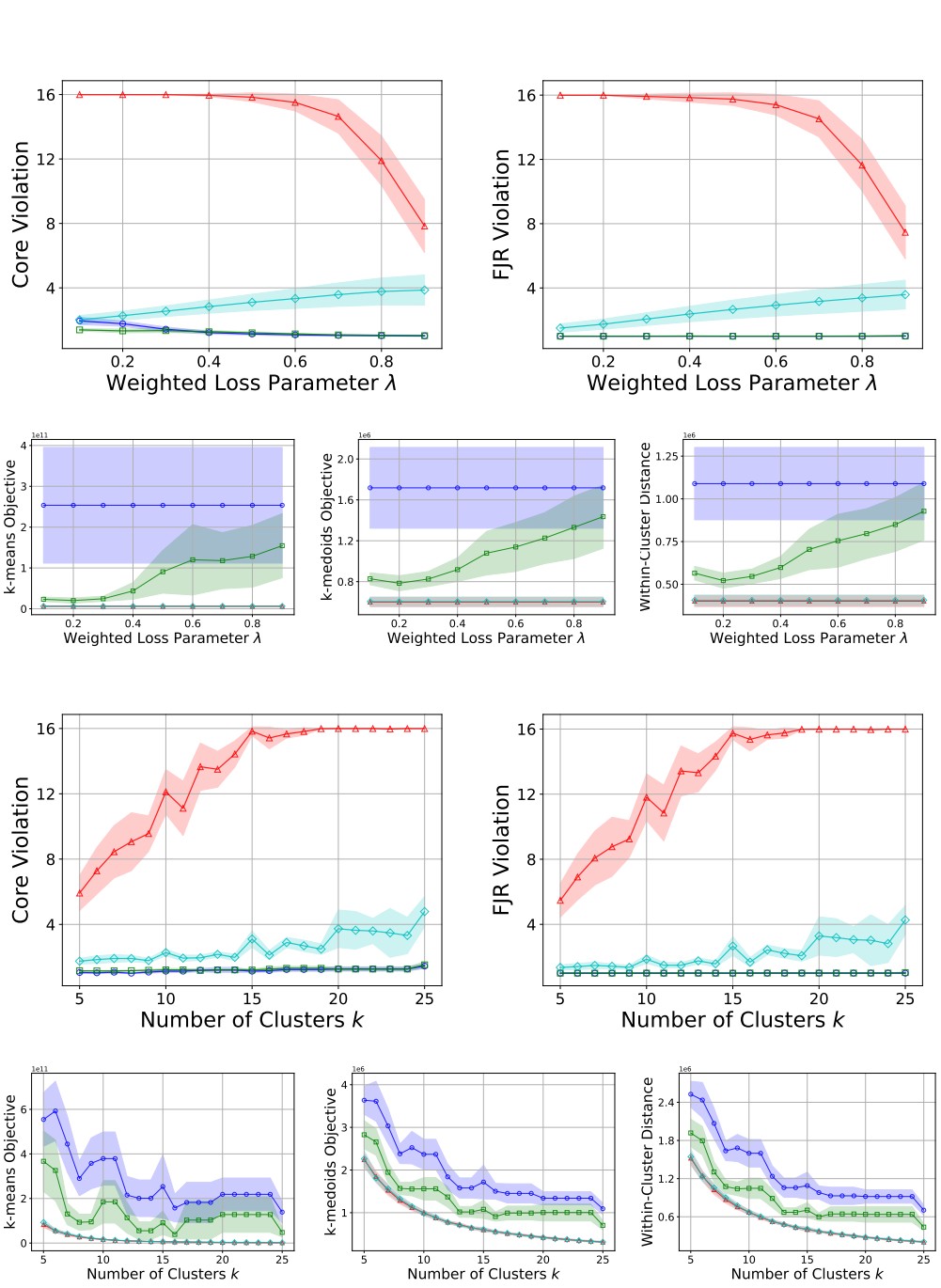

Figure 3: Results for the *Adult* (Census Income) dataset. The common legend appears at the top. The top five plots compare the algorithms on our five metrics when varying $\lambda \in \{0.1, 0.2, \ldots, 0.9\}$ and fixing $k = 15$, and the bottom five plots correspond to fixing $\lambda = 0.5$ and varying $k \in \{5, 6, \ldots, 25\}$.

exceeds a specified threshold, but here it is used for clustering the individuals. We apply a one-hot encoding to all categorical features.[5]

<hr/>

[5]Categorical features encoded as numerical values (e.g., $1, 2, 3, \ldots$) yield misleading distances in the absence of the one-hot encoding.

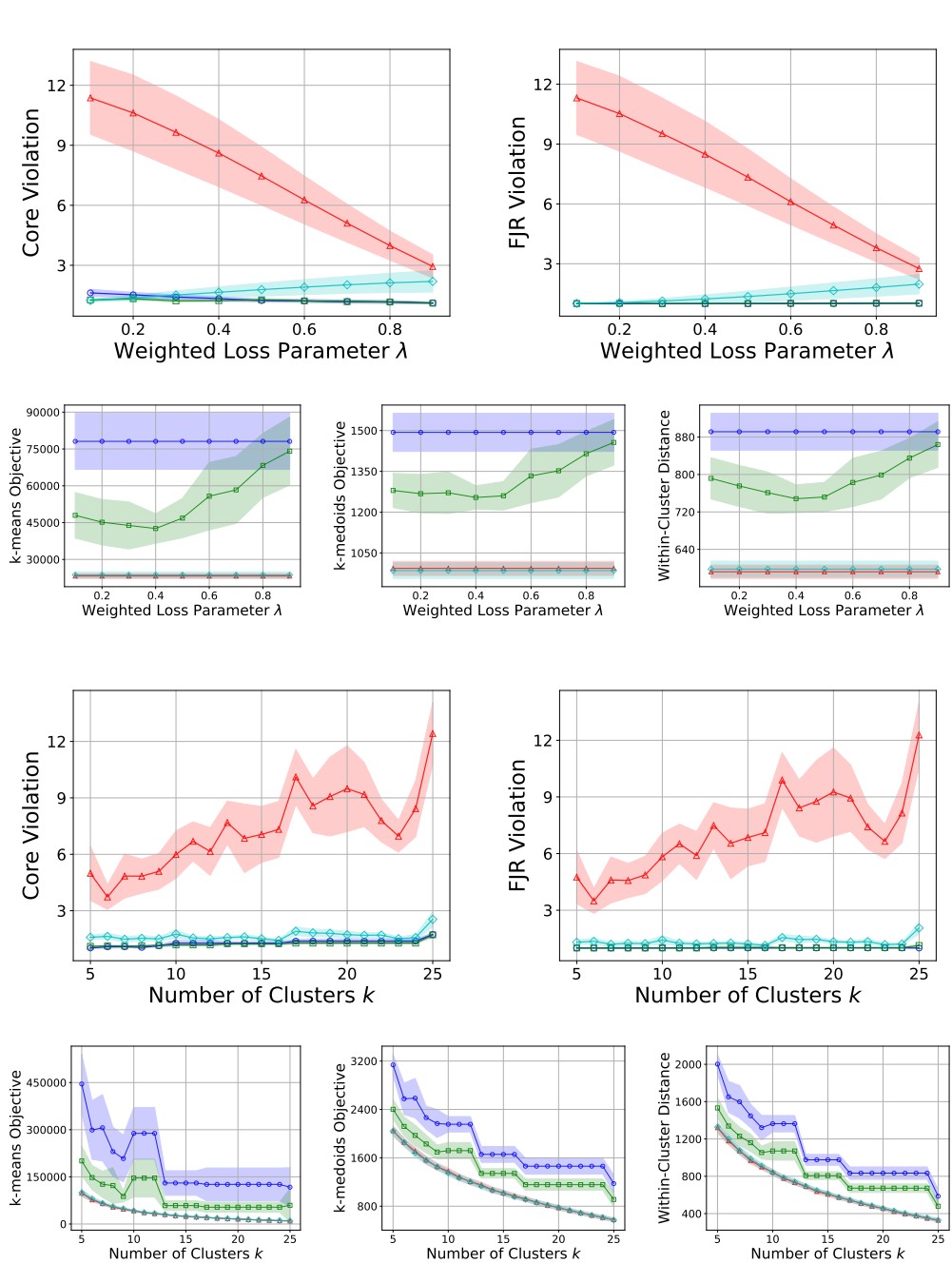

Figure 4: Results for the *Diabetes* dataset. The common legend appears at the top. The top five plots compare the algorithms on our five metrics when varying $\lambda \in \{0.1, 0.2, \ldots, 0.9\}$ and fixing $k = 15$, and the bottom five plots correspond to fixing $\lambda = 0.5$ and varying $k \in \{5, 6, \ldots, 25\}$.

**Metrics.** We evaluate the four algorithms above on five metrics.

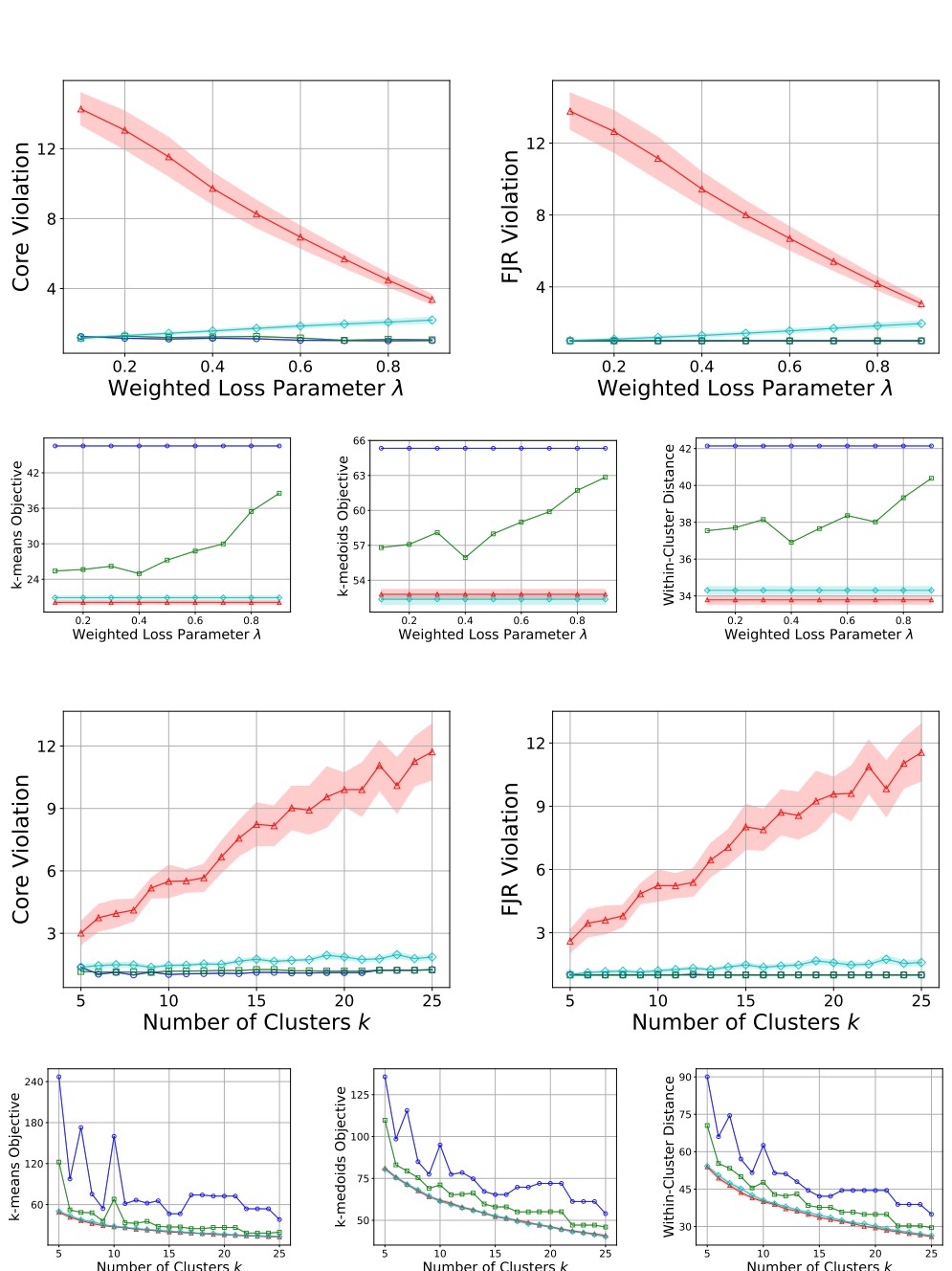

Figure 5: Results for the *Iris* dataset. The common legend appears at the top. The top five plots compare the algorithms on our five metrics when varying $\lambda \in \{0.1, 0.2, \dots, 0.9\}$ and fixing $k = 15$, and the bottom five plots correspond to fixing $\lambda = 0.5$ and varying $k \in \{5, 6, \dots, 25\}$.

The first two metrics are core and FJR violations (i.e., the smallest $\alpha$ such that the clustering is in $\alpha$-core or $\alpha$-FJR); these two are proportional fairness metrics. We compute the exact violations of core and FJR by solving integer linear programs, which, given a clustering, find a deviating cluster $(C, x)$ recording the largest possible violation of the corresponding metric.

The next three metrics are the $k$-means objective (stated above), the $k$-medoids objective (stated above), and average within-cluster distance, $\sum_{t\in[k]}\frac{1}{|C_t|}\sum_{i,j\in C_t} d(i,j)$ [30], which is another popular objective; these three can be considered efficiency metrics. These metrics are easy to compute given a clustering.

**Experimental setup.** Following Chen et al. [1], Caragiannis et al. [2], we assume $\mathcal{N} = \mathcal{M}$, and use the Euclidean $L_2$ distance metric. We vary two parameters: the number of clusters $k \in \{5, 6, \ldots, 25\}$ and the weighted loss parameter $\lambda \in \{0.1, 0.2, \ldots, 0.9\}$. Specifically, we report experimental results when varying $\lambda \in \{0.1, 0.2, \ldots, 0.9\}$ while fixing $k = 15$, and when varying $k \in \{5, 6, \ldots, 25\}$ while fixing $\lambda = 0.5$.

While our algorithms (and the traditional baselines) scale well, solving integer linear programs to compute the exact core and FJR violations of the clustering they return is computationally intensive. Hence, for the two larger datasets, namely *Adult* and *Diabetes*, we randomly sample 100 data points in each of 40 independent trials. For the *Iris* dataset, since $k$-means++ uses a randomized initialization step, we run it 20 times. Our plots show the five metrics above on average along with 95% confidence intervals.

**Results.** The results for *Adult*, *Diabetes*, and *Iris* datasets are presented in Figures 3 to 5, respectively. The qualitative takeaways are similar across the three datasets, so we use the *Adult* dataset (Figure 3) as an example to illustrate them.

First, our two algorithms, GC and SemiBall, achieve near-exact core and FJR, which is significantly better than their worst-case approximation guarantees from Lemmas 3 and 4. The classical algorithms, $k$-means++ and $k$-medoids, admit notable core and FJR violations, and their fairness deteriorates as $k$ increases; this may be because $k$ is still less than $n/2$, so an increase in $k$ increases the number of small coalitions (of size close to $n/k$) that can deviate. In particular, $k$-means++ is highly unfair, especially for larger values of $k$ and smaller values of $\lambda$ (i.e., when the centroid loss is dominant). Interestingly, $k$-medoids is reasonably fair; Caragiannis et al. [2] observe it to be unfair for the setting of $\lambda = 1$ (non-centroid loss), but our results show that its fairness improves as $\lambda$ decreases, and it becomes almost as fair as GC and SemiBall, producing near-exact core and FJR clustering, when $\lambda \to 0$ (centroid loss).

The increased fairness of GC and SemiBall comes at the cost of decreased efficiency metrics as compared to $k$-means++ and $k$-medoids. The efficiency loss is, however, small, especially when either $k$ is large or $\lambda$ is small. Among our two algorithms, SemiBall seems to consistently perform better in efficiency metrics (and equal in fairness metrics).

Hence, our main takeaway is that SemiBall and $k$-medoids are compelling algorithmic choices for semi-centroid clustering; the choice between the two may rely on the value of $k$ and $\lambda$, as well as the principal's desired fairness-efficiency tradeoff.

## G   Impossibility Results for Balanced Clustering

In this section, we provide two impossibility results for *balanced* clustering in our semi-centroid world. Informally, balancedness dictates that the partition of the set of agents contain clusters of roughly equal size. In many applications such as creating groups of students for class projects or designing oral presentation sessions at a conference, this may be a desirable criterion.

Formally, one may refer to a clustering $\mathcal{X} = \{(C_1, x_1), \ldots, (C_k, x_k)\}$ as *balanced* if $|C_t| \in \{\lfloor n/k \rfloor, \lceil n/k \rceil\}$ for all $t \in [k]$ (equivalently, $||C_t| - |C_{t'}|| \leqslant 1$ for all $t, t' \in [k]$). However, both our impossibility results hold even under the following weaker criterion.

**Definition 4** (Balanced Clustering). We call a clustering $\mathcal{X} = \{(C_1, x_1), \ldots, (C_k, x_k)\}$ *balanced* if, when $k$ divides $n$, $|C_t| = n/k$ for all $t \in [k]$.

It may be encouraging that non-centroid greedy capture indeed produces a balanced clustering under this weak version.[6] The centroid greedy capture, however, offers no such guarantees. This is because, while it still opens balls as soon as they capture $\lceil n/k \rceil$ uncaptured agents, (i) open balls continue to

---

[6]It may be possible that as soon as a ball covers at least $\lceil n/k \rceil$ uncaptured agents, it actually covers many more agents due to them being equidistant from the center. However, in this case, one can modify the algorithm to let the ball capture exactly $\lceil n/k \rceil$ of the uncaptured agents contained in the ball, leaving the rest on the boundary to

grow and capture more agents, and (ii) agents are reassigned to the nearest center in the end. At least the second aspect is retained to some degree in both our algorithm for the dual metric loss (Algorithm 1) and one of our algorithms for the weighted single metric loss (Algorithm 6). This makes the final sizes of the clusters rather unpredictable (and not necessarily roughly equal).

That said, Algorithm 5, which runs non-centroid greedy capture followed by a greedy centroid selection step, does in fact produce a balanced clustering and achieves $\frac{2}{\lambda}$-core with respect to the weighted single metric loss. Unfortunately, this becomes unbounded when $\lambda$ approaches 0 (the centroid world). The next result shows that this is impossible to avoid when requiring a balanced clustering. Hence, our use of Algorithm 6, which computes an imbalanced clustering to achieve a finite core approximation in the small $\lambda$ regime (thus yielding a finite core approximation across the entire spectrum of $\lambda \in [0, 1]$) is necessary.

**Theorem 10.** *For the weighted single metric loss with parameter $\lambda \in [0, 1]$, there exists an instance in which no balanced clustering is in the $\alpha$-core for any $\alpha < \frac{1}{\sqrt{\lambda}}$.*

*Proof.* Consider the instance from Figure 1. Note that $k = 3$ divides $n = 6$. Set $p = \frac{1}{\sqrt{\lambda}}$. For contradiction, assume that there exists a balanced clustering $\mathcal{X} = \{(C_1, x_1), (C_2, x_2), (C_3, x_3)\}$ with $|C_1| = |C_2| = |C_3| = 2$, which is in the $\alpha$-core for some $\alpha < \frac{1}{\sqrt{\lambda}}$.

First, note that $\{a, b\}$ must be one of the clusters. Suppose for contradiction that this is not the case. Then, at most one of agents $a$ and $b$ can be clustered with $c$ (i.e., at most one of $\mathcal{X}(a) = \mathcal{X}(c)$ and $\mathcal{X}(b) = \mathcal{X}(c)$ is true). Without loss of generality, suppose that agent $a$ is clustered with $c$ (a symmetric argument holds when agent $b$ is clustered with $c$). Then, $\ell_a(\mathcal{X}) \geqslant \lambda \cdot p = \sqrt{\lambda}$; this lower bound holds due to the non-centroid loss part alone. Since agent $b$ is clustered with an agent in $\{d, e, f\}$ that it is infinitely far from, $\ell_b(\mathcal{X}) = \infty$. Then, consider the deviating cluster $(\{a, b\}, a)$, which yields $\ell_a(\{a, b\}, a) = \lambda$ and $\ell_b(\{a, b\}, a) = 1$. Compared to $\mathcal{X}$, agent $b$ improves its loss by an infinite factor and agent $a$ improves its loss by a factor of $\frac{\sqrt{\lambda}}{\lambda} = \frac{1}{\sqrt{\lambda}}$, which contradicts the fact that $\mathcal{X}$ is in the $\alpha$-core for some $\alpha < \frac{1}{\sqrt{\lambda}}$.

Therefore, $\{a, b\}$ must be one of the clusters and at least one of agents $a$ and $b$ is not the centroid of this cluster. Without loss of generality, assume that the centroid is not at agent $b$. Then, consider the deviating cluster $(\{b, c\}, b)$. $\ell_b(\{b, c\}, b) = \lambda \cdot p = \sqrt{\lambda}$ and $\ell_c(\{b, c\}, b) = p$. In contrast, $\ell_b(\mathcal{X}) \geqslant 1$, so agent $b$ improves by a factor of at least $\frac{1}{\sqrt{\lambda}}$. Further, since $\{a, b\}$ is a cluster, agent $c$ must be clustered with an agent from $\{d, e, f\}$ that it is infinitely far from, yielding $\ell_c(\mathcal{X}) = \infty$. This means agent $c$ improves by an infinite factor. This contradicts the fact that $\mathcal{X}$ is in the $\alpha$-core for some $\alpha < \frac{1}{\sqrt{\lambda}}$. This completes the proof. $\square$

Since the broader class of dual metric loss contains the weighted single metric loss for all $\lambda \in [0, 1]$, it follows that one cannot hope to design a balanced clustering algorithm that achieves any finite approximation to the core with respect to the dual metric loss.

**Theorem 11.** *Under the dual metric loss, no algorithm that always produces a balanced clustering can achieve $\alpha$-core for any $\alpha \geqslant 1$.*

*Proof.* Suppose for contradiction that there exists an algorithm that always produces a balanced clustering in the $\alpha$-core with respect to the dual metric loss. Then, consider the instance from Figure 1, and set the non-centroid metric as $d^m = \lambda \cdot d$ and the centroid metric as $d^c = (1 - \lambda) \cdot d$ for any $\lambda < 1/\alpha^2$. The induced dual metric loss is $\ell_i(C, x) = \lambda \cdot \max_{j \in C} d(i, j) + (1 - \lambda) \cdot d(i, x)$, which is precisely the weighted single metric loss induced by distance metric $d$ and parameter $\lambda$. From Theorem 10, the core approximation achieved by any balanced clustering, including the one produced by the algorithm, must be at least $\frac{1}{\sqrt{\lambda}} > \alpha$, which is the desired contradiction. $\square$

---

be captured by other balls in the future. This modification does not affect any of the guarantees achieved by the algorithm. Further, it produces a balanced clustering as per Definition 4. Note that when $k$ does *not* divide $n$, the last non-empty cluster may contain fewer than even $\lfloor n/k \rfloor$ agents.

