# OpenReview forum: "Unifying Proportional Fairness in Centroid and Non-Centroid Clustering"
_NeurIPS.cc/2025/Conference — NeurIPS 2025 spotlight_

### Official Review · Reviewer_ZM2J · 2025-06-26

**Clarity:** 3
**Significance:** 3
**Originality:** 3
**Rating:** 5
**Confidence:** 3

**Summary:**

The paper studies proportional clustering. In particular, it studies a setting generalizing both centroid and non-centroid clustering. That is, the goal is to partition a set of points (like in non-centroid clustering) and assign each partition to a center (like in centroid clustering) at the same time, with the loss of each agent being a combination of its centroid and non-centroid loss. The objective studied in the paper is proportional fairness. That is, ideally, with n points and k cluster centers approximately n/k points should deserve their own partition and center. The paper introduces several definitions of proportional fairness (depending on the loss function of the points and the strength of the deviating coalitions) and gives algorithms and lower bounds for achieving these proportional fairness definitions.

**Questions:**

I would be interested to hear answers to the two questions I had in my weaknesses section.

1. Are you aware of any other papers studying this setting or a related setting where one might need both centroid and non-centroid clustering?
2. In the setting in which N = M, for a given cluster (C, x) the cluster center x must not necessarily be a part of C (and this is also not enforced by the algorithms as far as I can see). Do the results still hold if this is enforced or might this not even be desirable?

With sufficient answer to these questions (in particular the first question) I am willing to upgrade to an accept.

**Ethical Concerns:**

["NO or VERY MINOR ethics concerns only"]

**Final Justification:**

Solid paper, interesting theory (also interesting experiments, albeit a bit hidden, but I trust the authors to highlight this better)

**Limitations:**

Yes

**Quality:**

3

**Strengths And Weaknesses:**

Strengths: Proportional fairness is an interesting topic and this paper makes a novel contribution to it that should be of interest to the researchers working on proportional fair clustering. In fact, I have  asked myself something similar akin to Theorem 1 while reading through the paper of Caragiannis et al. and am happy that there is a nice simple (albeit negative) solution to this. I also find the study of combining centroid and non-centroid clustering to be somewhat interesting and novel. The algorithms presented throughout the paper seem are for the most part non-trivial and clearly presented, and I think the authors largely do a good job at making the paper readable. The results also seem technically interesting, and I think the two-phase Algorithm 1 is a nice twist on the usual Greedy Capture based algorithms.

Weaknesses:

I think one obvious weakness of the paper is the setting. In particular (while seemingly novel), I find the study of both centroid and non-centroid clustering at the same time to be slightly unnatural, and I think it could have been better motivated in the paper. Are you aware of any other papers studying this setting or a related setting where one might need both centroid and non-centroid clustering?

Secondly, one thing I found a bit unnatural, is that in the setting in which N = M, for a given cluster (C, x) the cluster center x must not necessarily be a part of C (and this is also not enforced by the algorithms as far as I can see). Do the results still hold if this is enforced or might this not even be desirable?

The presentation of the paper could be improved at times. In particular, I think the bounds presented in Lemma 4, Theorem 5, Theorem 6 should be explained. For further things, see detailed comments later.

Finally, while I usually think that a theory paper does not need experiments, I think they would be nice. In fact, the paper even has experiments and I would have loved to see them in the main text. In particular, I would have been interested to see whether the worst case presented in Theorem 1 actually happens in real world instances.

Detailed Comments:
-The equivalence between the two forms of k-means presented at the start of the paper do not hold for the setting here, in which there is separate candidate set.
- “which is a principle at the heart of our democratic institutions.“ I am not a fan of this phrasing. Proportionality is not even at the heart of American institutions.
-I find Table 1 to be quite unreadable without context. In particular, I think that every variable in a table should be introduced to make the table understandable.
-”approximation to the core” without context, it is not known what kind of “approximation” this is supposed to be
- For stylistic purposes, the order of the losses should be the same in lines 129 and 136
- I think sorted references are easier to access for the reader. Also, you currently have abbreviated first names for some, and spelled out for others, papers 13 and 17 are no longer forthcoming, and paper 28 has a conference version.

---

> ### Author Rebuttal · Authors · 2025-07-31
>
> Thank you for your detailed and helpful review.
>
> **Comments:**
>
> * *Equivalence between two forms of k-means*: We agree that this phrasing was misleading. We meant to provide only an example setting (where $\mathcal{M}$ is the entire Euclidean space and $d = L_2$) in which the "centroid" and "non-centroid" k-means become equivalent. We will clarify this detail in the camera ready version.
>
> * *Regarding proportionality principle wording:* We agree with your comment and will revise this as: "which is a principle inspired from the democratic ideal of proportional representation that aims to give groups influence proportional to their size and cohesion."
>
> * We will also make the changes to the references and the stylistic changes you suggested.
>
> * *Regarding experiments:* As we mentioned to Reviewer wHCe, we will update Section 1.1 in the camera-ready version to include a more detailed synopsis of our empirical findings.
>
> **Questions:**
>
> **Q1: Motivation for semi-centroid clustering**: Please see our responses to Reviewer wHCe's Q1, SNW3's Q1, and sbW8's Q3. In Section 6, we discuss motivating applications such as assigning students to group projects and scheduling talks of papers accepted to a conference, where both non-centroid and centroid losses matter simultaneously.
>
> Quoting from our response to Reviewer SNW3:
>
> > One such application is assigning students to groups for a class project. In this setting, the agents would be the students, and the centroids would be project topics. The non-centroid metric space would represent the friendships of the students, while the centroid metric would represent students' preferences over project topics. It may be impossible to embed both these preferences in a single metric space, as we may have two students who are very close friends, but have very different preferences over project topics. The dual metric model allows us to take this into account and find a grouping of the students that balances placing friends in the same group with assigning students to topics of their liking.
>
> For scheduling talks of papers accepted to a conference, the non-centroid loss would model the (dis)similarity between two papers, while the centroid loss would model the (ir)relevance of a session's title to a paper in that session.
>
> While there are several real-world applications where agents are grouped into clusters and each cluster is assigned a "centroid", coming to your question, we are not aware of any technical papers or settings studied in prior work that model both the non-centroid and centroid aspects simultaneously. This speaks to the conceptual novelty of our work.
>
> **Q2: The case of $\mathcal{M}=\mathcal{N}$.** This is an extremely interesting question. Due to space constraints, we were only able to allude to this special case in the Discussion (see line 366).
>
> For the dual metric setting, Algorithm 1 with $\mathcal{M}=\mathcal{N}$ indeed does not guarantee that the centroid for each cluster will be an agent from that cluster because the MCC computed at a step may select a centroid from outside the cluster. It is easy to see that this is inevitable if one wishes to prevent clusters of the form $(C,x)$ with $x \notin C$ from deviating and significantly improving. However, if the desideratum is weakened to approximate core against only deviating clusters $(C,x)$ that satisfy $x \in C$, we do not know if it can be achieved for the dual metric setting.
>
> However, whenever $\mathcal{M} = \mathcal{N}$, it is presumable that both centroid and non-centroid losses are represented by the same metric, meaning that we are in the case of weighted single metric loss. Here, it is worth noting that one of our algorithms, Algorithm 5, naturally achieves your desideratum: Algorithm 5 simply runs non-centroid GreedyCapture (where balls grow around the agents) and then assigns each cluster the centroid closest to the agent $i \in C$ at the center of the ball that captured the cluster. When $\mathcal{M} = \mathcal{N}$, we are guaranteed to have $x=i$ WLOG, meaning that the centroid will always be an agent in the cluster. Algorithm 5 gets a core approximation of $\frac{2}{\lambda}$, which is reasonable for all lambda values that are not too close to 0.
>
> That said, Algorithm 6, which allows us to achieve finite core approximation for the regime of $\lambda$ close to $0$ under weighted single metric loss, does not always select a centroid from within the cluster. Whether this can be enforced remains an interesting open question.
>
> We will add this discussion to Section 6.

---

> > ### Comment · Reviewer_ZM2J · 2025-08-04
> >
> > Thank you for the detailed response! While I am still not 100% convinced of the motivation, I think the paper's theoretical contribution is strong enough to counteract this, and I will upgrade the rating.

---

### Official Review · Reviewer_sbW8 · 2025-07-02

**Clarity:** 3
**Significance:** 2
**Originality:** 2
**Rating:** 5
**Confidence:** 5

**Summary:**

Motivated by the fact that the main algorithms for proportional non-centroid and centroid clustering (called Greedy Capture) look almost the same,
the authors consider a clustering variant where one simultaneously computes a non-centroid and a centroid clustering, for possibly separate metrics.
The resulting problems partitions the given points into k clusters and maps a center point to each of these clusters.
The goal is to find such a clustering that is (approximately) proportional simultaneously for the centroid-based and the non-centroid based loss function, i.e.,
the distance to the center and the maximum distance to any other point in the same cluster.
The authors present a modification of the Greedy Capture algorithm, where one first builds a tentative cluster and then allows some agents to swap clusters and makes use of a subroutine that is NP-hard to compute, but efficiently approximable.
Assuming access to optimal solutions to the subroutine, the authors show that there always exists a clustering in the 3-core,
and using the approximate subroutine, the authors show that one can efficiently compute a clustering in the (3+2sqrt(3))-core.

The authors further show some results for the case that centroid and non-centroid clustering use the same metric and also consider a relaxation of the core notion called FJR.
These results are based mainly on small variations of existing algorithms.

**Questions:**

- What is the best factor alpha obtainable by Algorithm 1 when only focusing only on the centroid loss or only on the non-centroid loss?
- Suppose that you are given a non-centroid clustering in the alpha-non-centroid core. Is it possible to extend this to a semi-centroid clustering that is in the alpha*beta-core for some constant beta?

- I suggest that you slightly extend on the motivation for federated learning. It seems pretty plausible, but needs to be more detailed if the greater audience should be able to follow.

**Ethical Concerns:**

["NO or VERY MINOR ethics concerns only"]

**Final Justification:**

I increase my score mainly due to a slightly new perspective on the paper.
The idea of combining the non-centroid and centroid versions of Greedy Capture itself already deserves some credit: While maybe a theoretical thought process, it nicely ties the two directions within proportional clustering together. It now just happens that this also works for two different metrics.

**Limitations:**

yes

**Quality:**

3

**Strengths And Weaknesses:**

Strengths
- From a theoretical perspective, the authors approach the intriguing question why almost the same algorithm works for two quite different variants of clustering (albeit with similar fairness notions) and find a satisfying answer: For optimizing both notions at the same time, one needs to deviate a bit further from the Greedy Capture approach. The practical motivation via federated learning also seems plausible in the sense that, if I would rather deviate to another model, then I also need to find a group of peers who would join me in the deviation.
- The paper presents itself very well, the proofs are clearly written and easy to verify.

Weaknesses
- While the theoretical motivation is very strong, the practical motivation is a bit faltering to me. I do believe that the federated learning motivation could somehow be interesting, and I like the idea of having the centroids represent a model. I do believe that it would help add more details to this motivation.
- Overall, the technical depth of the work is not too high. The new Algorithm 1 is a nice modification from Greedy Capture, but still lends many elements from the algorithms by Chen et al. and Caragiannis et al., such as the most cohesive cluster subroutine.
- The impact of the work is not too high. In all, the work seems a bit niche and incremental.

---

> ### Author Rebuttal · Authors · 2025-07-31
>
> Thank you for your detailed review.
>
> **Q1: Best factor for centroid or non-centroid clustering.** The approximation guarantees of Algorithm 1 are closely controlled by the growth factor $c$. Recall that $c$ dictates how much an agent $i$ is allowed to hurt the losses of other agents when switching into their cluster for its own benefit during Phase 2 of the algorithm.
>
> Theorem 2 indicates that setting $c=3/2$ can already recover $3$-core guarantee for each of centroid and non-centroid clustering (by setting the other distance metric as $0$). However, setting different values of $c$ can recover optimized guarantees in each setting from prior work.
>
> Specifically, when we set $c=0$, agents will not be allowed to switch away from their original cluster if it hurts the agents in that cluster at all. This essentially means that Phase 2 of the algorithm will be skipped, and Algorithm 1 will roughly mimic the behaviour of the non-centroid GreedyCapture of Caragiannis et al. [2024] to achieve $2$-core for non-centroid clustering. Similarly, if we set $c=\infty$ (formally, if in Phase 2 we say that every agent can simply switch to its favourite cluster, regardless of the loss it causes other agents), then the algorithm would mimic the centroid GreedyCapture of Chen et al. [2019] to achieve $(1+\sqrt{2})$-core for centroid clustering.
>
> With these respective growth factors, the only difference between our Phase 1 and the GreedyCapture algorithms for these settings is that Algorithm 1 finds the Most Cohesive Cluster at each step rather than finding the clusters by growing a ball around each agent or cluster center. By analyzing the original greedy capture proofs in both settings, it is easy to see this difference does not impact any of the guarantees.
>
> **Q2: Extending non-centroid clustering to semi-centroid clustering.** That's a great question. The answer is no. This follows from the fact that the non-centroid GreedyCapture of Caragiannis et al. [2024] computes a balanced clustering (where each cluster is of size at most $\lceil n/k \rceil$) in the 2-core (i.e., $\alpha=2$ in your question). However, in Appendix F, we show that there are instances of semi-centroid clustering in which every balanced clustering has core approximation at least $1/\sqrt{\lambda}$. Hence, when $\lambda = o(1)$, one cannot guarantee a constant $\beta$ in your question.
>
> **Q3: Motivation**: Indeed, we will expand our federated learning motivation to explain the setup more in detail. That said, our goal was to demonstrate that the motivations used by prior work for centroid and non-centroid clustering may already involve *at least a slight* semi-centroid consideration. We discuss new motivating examples in Section 6 that more directly align with semi-centroid clustering, which we plan to discuss early in Section 1 in the camera-ready version (see also our responses to Reviewer wHCe's Q1 and SNW3's Q1).

---

> > ### Comment · Reviewer_sbW8 · 2025-08-05
> >
> > Thank you for the response. I am still not quite convinced by the motivation, and I think it would be great if the authors expand on this in the camera-ready version. I do like the theoretical contributions very much; I will increase my score to reflect this.

---

### Official Review · Reviewer_SNW3 · 2025-07-03

**Clarity:** 3
**Significance:** 3
**Originality:** 2
**Rating:** 5
**Confidence:** 3

**Summary:**

In this paper the authors explore two notions of fairness in a clustering model that lies between centroid and non-centroid clustering. The task is to partition a set of $n$ data points (agents) into $k$ clusters, each defined by a center point and a group of assigned agents. An agents loss is defined as the sum of the distance to its assigned center and the maximum distance to any other agent within the same cluster. A clustering is considered unfair if there exists a group of $\frac{n}{k}$ agents who could form their own cluster in which each member would experience a (by some amount) lower loss.

The authors analyze two fairness notions—the 'core' and 'fully justified representation'— and provide both upper and lower bounds on their approximation. They distinguish between two settings: one where the distance towards the center and the distance to other agents are measured using different metrics and another where the two metrics are identical, up to a constant factor. For both settings, the paper presents efficient algorithms that achieve constant-factor approximations of the fairness notions.

**Questions:**

What is the motivation for allowing two different metrics in the loss function? Why would an agent measure the distance to the center differently than the distance to the other agents?

**Ethical Concerns:**

["NO or VERY MINOR ethics concerns only"]

**Final Justification:**

My questions were nicely answered. While they did not fully resolve my slight concerns about the practical relevance of the topic, I still recommend acceptance.

**Limitations:**

yes

**Paper Formatting Concerns:**

/

**Quality:**

4

**Strengths And Weaknesses:**

The authors approach a natural question: Given existing fairness results for both centroid and non-centroid clustering, what if an agent cares about both being close to the center as well as to other members of their cluster? While the application-based motivation is somewhat weak, the problem itself is interesting and bridges the gap between different approaches in the literature.
The paper is well written and gives a clear overview of prior related work. The results seem significant and contribute to the understanding of fairness in clustering. With most of the proofs in the appendix, the authors generally succeed in conveying the key ideas behind their results and algorithms. That said, Theorem 1 would benefit from including the example instance from the proof in the main body. For Theorem 4 and 5 (or result summary table) a figure plotting the upper and lower bounds as a function of $\lambda$ would greatly enhance clarity and help readers to interpret the bounds and the gap between them.

Typo in line 226: with respect the -> with respect to the

---

> ### Author Rebuttal · Authors · 2025-07-31
>
> Thank you for your review.
>
> **Regarding the example in Theorem 1:** Indeed, the counterexample used to prove Theorem 1 is insightful as we later reuse it to prove Theorems 5 and 9 too. We'll be happy to add it to the main body.
>
> **Plot of upper and lower bounds:** This is a great idea. We will be happy to add such a plot to the main body.
>
> **Q1: Motivation for two different metrics.** There are several scenarios where having different metrics can be helpful. We briefly describe the motivation in Section 6, but agree that this belongs to the introduction and will move it there in the camera-ready version.
>
> One such application is assigning students to groups for a class project. In this setting, the agents would be the students, and the centroids would be project topics. The non-centroid metric space would represent the friendships of the students, while the centroid metric would represent students' preferences over project topics. It may be impossible to embed both these preferences in a single metric space, as we may have two students who are very close friends, but have very different preferences over project topics. The dual metric model allows us to take this into account and find a grouping of the students that balances placing friends in the same group with assigning students to topics of their liking.

---

> > ### Comment · Reviewer_SNW3 · 2025-08-02
> >
> > Thank you for your answers. I have no further questions at this point.

---

### Official Review · Reviewer_wHCe · 2025-07-05

**Clarity:** 4
**Significance:** 3
**Originality:** 3
**Rating:** 5
**Confidence:** 2

**Summary:**

This paper introduces and investigates a generalized clustering framework called semi-centroid clustering, which unifies the two previously studied paradigms in proportionally fair clustering: centroid-based clustering (where loss is defined by distance to a cluster centroid) and non-centroid clustering (where loss depends on distances to other agents in the cluster). By simply combining the two associated cost functions under different settings, the authors initiate study on two important concepts for clustering, the core and fully justified representation (FJR). The major contribution in balancing these two clustering notions is a non-trivial polynomial-time constant approximation algorithm for the core under different loss functions for the two (simultaneous) objectives.

**Questions:**

How should a practitioner choose between different values of $\lambda$ in a real application? How sensitive are the algorithms to the choice of metric or to approximation factors in MCC computation?

How do these algorithms scale in practice / what is their runtime guarantee?

How might general loss formulations might be captured in future work?

**Ethical Concerns:**

["NO or VERY MINOR ethics concerns only"]

**Limitations:**

While the paper acknowledges that simultaneous core approximation is infeasible, it would be valuable to better discuss practical implications

**Quality:**

4

**Strengths And Weaknesses:**

STRENGTHS

This paper makes a significant theoretical contribution by bridging two previously distinct lines of research in proportionally fair clustering.
The dual and weighted metric loss models serve as natural and flexible interpolants between existing paradigms, readily allowing extension beyond the two main directions of clustering algorithm design.
The technical results are nontrivial, with the core approximation algorithm under dual metric loss is a highlight.
In the weighted single metric loss setting, the authors provide a nuanced analysis with a full picture of the approximation frontier as parametrized by a trade-off.
The authors further validate their problem definition and algorithm with experimental results, though these are entirely deferred to the appendix.

WEAKNESSES

My main concern is that, while the results and framing are very interesting, the complete paper is largely contained in the appendix and suggests the work may actually be better situated for a more theory heavy conference.
A short section in the main body summarizing key empirical findings would improve accessibility and demonstrate the algorithms’ relevance to real-world clustering scenarios.
The paper focuses on linear interpolation between the two cost metrics. The work does not explore alternatives to these, or discuss more expressive models

The one writing weakness I found is that the algorithm descriptions are a bit hard to parse for a non-expert in the field. If possible (with a longer page limit) more intuitive examples or figures to illustrate algorithm behavior would be beneficial to the reader.


MINOR FIXES:
- some inconsistency in the text with "semi centroid" vs "semi-centroid"
- the presentation of Algorithm 1 is very cumbersome with notation
- no figures or presentation of experimental results in main text

---

> ### Author Rebuttal · Authors · 2025-07-31
>
> Thank you for your detailed and helpful review.
>
> We will fix the typos you found and simplify the presentation of Algorithm 1 in the camera-ready version of the paper.
>
> **Regarding experiments:** While we would have liked to include them in the main body, we were restricted by the page length. In the camera-ready version, we will significantly expand on the paragraph in Section 1.1 that mentions the experiments to include a summary of our key findings.
>
> **To address your questions:**
>
> **Q1: Choosing $\lambda$, metric, and MCC approximation.**
>
> *Choosing $\lambda$:* In real-world scenarios, the correct value of $\lambda$ would likely be very context dependent. For example, when clustering conference papers into sessions, it is presumable that the non-centroid loss (similarity between papers presented in the same session) matters significantly more than the centroid loss (relevance of a session's title to a paper in that session), leading the conference chair to select $\lambda$ close to $1$. However, when clustering students for group projects, the centroid loss (how much a student prefers to work on a given project idea) may matter just as much as the non-centroid loss (how much two students like working with each other), leading the educator to select a more intermediate value of $\lambda$. In general, the practitioner would use their insight as to which factor matters more (or if they both matter equally) and choose $\lambda$ to reflect that. (See also our response to Reviewer SNW3's question.)
>
> *Choosing the metric(s):* There are two ways to understand your comment about sensitivity to the choice of the metric(s).
>
> 1. *What if a practitioner feeds slightly distorted metrics $\hat{d}^c$ and $\hat{d}^m$ that are at most a $1+\epsilon$ (multiplicative) factor off from the true $d^c$ and $d^m$ in measuring any pairwise distance?* In this case, we have checked that all our approximation guarantees scale gracefully in $\epsilon$. The distortion factor just carries through all the equations. If the reviewers feel that this would be helpful, we will be happy to add formal robustness results in the appendix of the camera-ready version.
>
> 2. *Could the approximation factors improve for specific choices of $d^c$ and $d^m$?* Prior work of Micha and Shah [2020] shows that for centroid clustering, the core approximation of GreedyCapture improves under the $L_2$ distance in a Euclidean space. It would not be surprising if the same holds in our semi-centroid setting. This would require novel arguments and is left to future work.
>
> *Choosing MCC approximation factor:* For our algorithms, an $\alpha$ approximation of MCC smoothly translates to an $\alpha$ approximation of FJR and a $\frac{1}{2}(1+\sqrt{\alpha(\alpha+8)}+2)$ approximation of the core (see Theorem 6 and Lemma 1, respectively). That said, a practitioner would likely only choose one of two values: $\alpha=1$, if running time is not a concern and the practitioner is willing to solve a simple MILP with $O(|N|+|M|)$ variables and constraints to find a $1$-MCC cluster, and $\alpha=4$, if polynomial running time is a concern and the practitioner wishes to use Algorithm 7.
>
> **Q2: running time.** With $n=|N|$ and $m=|M|$ (and noting that $k \le n$), a simple implementation of GreedyCapture runs in $O(n m \log (nm))$ time. Algorithm 4 for computing a $4$-MCC cluster runs GreedyCapture $m$ times, meaning Phase 1 of Algorithm 1 runs in $O(n m^2 \log (nm))$ time. Phase 2 of Algorithm 1 runs in $O(n^2)$ time. This puts the overall runtime of Algorithm 1 at $O(n m^2 \log (nm))$. Our algorithms for the single-metric setting have similar runtimes. Algorithms 5 and 6 both rely on greedy capture and can be implemented in $O(n^2 \log n + km)$ and $O(n m \log (nm) + kn)$ time, respectively. We will add these details in the camera-ready version.
>
> Empirically, we observed that our algorithms are slower than k-means and k-medoids. As an example, the table below shows the runtimes (in seconds) of Algorithm 6, k-means and k-medoids on the Adult dataset subsampled to different sizes, fixing the default values of $k=15$ and $\lambda=0.5$. Please see the Appendix E for details of the experimental setup and implementation. Designing faster (perhaps even near-linear time) algorithms with provable proportional representation guarantees remains an exciting open question for future research.
>
> | # Points | Alg 6   | k-means | k-medoids |
> | -------- | ------- | ------- | -------   |
> | 800      | 0.254s  | 0.137s  | 0.025s    |
> | 6400     | 13.75s  | 1.155s  | 3.283s    |
> | 25600    | 291.25s | 3.401s  | 128.85s   |
>
> **Q3: general losses.** For arbitrary loss functions, the core remains inapproximable to any finite degree, even in non-centroid clustering [Caragiannis et al., 2024], whereas (exact) FJR remains feasible (Theorem 6). An interesting future direction would be to find families of loss functions that are more general than our dual metric loss, but still allow a reasonable approximation of the core.

---

### Author Response · Authors · 2025-08-09

Thank you again to all the reviewers for your thoughtful feedback and for reading our responses. We are especially grateful for the comments pointing out typos and for the suggestions to improve clarity. We will be incorporating all these suggestions into the final version of the paper.

---

### Decision · Program_Chairs · 2025-09-17

**Decision:**

Accept (spotlight)

**Comment:**

The paper unifies two versions of an algorithm called greedy capture for proportionally fair clustering. Fair clustering is a big trend and the submission is very interesting. While the reviewers are not fully convinced of the practical motivation, the technical contribution convinced all reviewers and the area chair to recommend acceptance of the paper.